# D-CIPHER: Discovery of Closed-form Partial Differential Equations

**Krzysztof Kacprzyk**
University of Cambridge
kk751@cam.ac.uk

**Zhaozhi Qian**
University of Cambridge
zq224@cam.ac.uk

**Mihaela van der Schaar**
University of Cambridge,
The Alan Turing Institute
mv472@cam.ac.uk

## Abstract

Closed-form differential equations, including partial differential equations and higher-order ordinary differential equations, are one of the most important tools used by scientists to model and better understand natural phenomena. Discovering these equations directly from data is challenging because it requires modeling relationships between various derivatives that are not observed in the data (*equation-data mismatch*) and it involves searching across a huge space of possible equations. Current approaches make strong assumptions about the form of the equation and thus fail to discover many well-known phenomena. Moreover, many of them resolve the equation-data mismatch by estimating the derivatives, which makes them inadequate for noisy and infrequent observations. To this end, we propose D-CIPHER, which is robust to measurement artifacts and can uncover a new and very general class of differential equations. We further design a novel optimization procedure, CoLLie, to help D-CIPHER search through this class efficiently. Finally, we demonstrate empirically that it can discover many well-known equations that are beyond the capabilities of current methods.

## 1 Introduction

Scientists have been using mathematical equations to describe the world for centuries. In particular, closed-form differential equations turned out to be one of the best tools to model physical phenomena. A differential equation describes a relationship between a quantity and its derivatives (rates of change); it is called closed-form if this relationship is described by a mathematical expression consisting of a finite number of variables, constants, arithmetic operations, and some well-known functions (e.g., exponent, logarithm, trigonometric functions)[1]. Closed-form differential equations provide a general description of reality in a concise representation that is amenable to closer inspection by scientists. This renders them transparent and interpretable to human experts.

Discoveries of these equations required a thorough knowledge of the theory, strong mathematical skills, substantial creativity, and good intuition. The goal of this work is to discover closed-form differential equations directly from data thus accelerating the process of scientific discovery.

**Challenges in discovering differential equations from data**

- **Partial and higher-order derivatives.** Many algorithms [10, 42] can only identify Ordinary Differential Equations (ODEs) which evolve only with respect to one variable (usually time). In contrast, many natural phenomena are described by equations involving many variables (e.g., spatial coordinates) called Partial Differential Equations (PDEs). Many equations also involve higher-order derivatives.

---

[1]Detailed discussion in Appendix A.2

- **Derivatives not observed.** Discovering differential equations from data is challenging because the derivatives are usually not observed in the dataset (*equation-data mismatch* [42]). This makes verifying a candidate equation a non-trivial task. Most of the methods try to resolve this issue by estimating the derivatives [10, 47]. However, derivative estimation is difficult, especially when the data is sampled infrequently or with high noise [42, 35]. For an illustrative study, see Appendix F.
- **Strong assumptions and constrained search space.** The majority of algorithms for identifying differential equations make many assumptions about the form of the equation. In particular, they make the *evolution assumption* (defined and explained later) and assume that the equation can be represented as a linear combination of prespecified functions and differential operators [10, 35]. However, many well-known equations, such as a forced harmonic oscillator or an inhomogeneous wave equation, cannot be represented in that way.

Currently, a few algorithms tackle only some of these challenges. In particular, Weak SINDy [35] is able to discover PDEs without estimating the derivative by utilizing a variational approach. However, the form of the equation is constrained to be in a form amenable for a sparse regression algorithm. D-CODE [42], on the other hand, uses a variational approach in conjunction with a symbolic regression algorithm to discover closed-form ODEs. However, it cannot handle higher-order derivatives or multiple independent variables, so it cannot be used to discover closed-form PDEs. The algorithms that do not require the evolution assumption appeared in [33] and [23] but they require derivative estimation and only consider equations represented as linear combinations of prespecified functions.

**Contributions.** In this work, we develop the **D**iscovery of **C**losed-form **P**artial and **H**igher-order Differential **E**quations in a **R**obust Way framework (D-CIPHER) that does not estimate the derivatives, requires fewer assumptions, has a broader search space than previous techniques, and works for both higher-order ODEs and PDEs. Our contributions are as follows:

- We examine the landscape of different types of PDEs from the discovery perspective. In particular, we introduce new notions such as *evolution form*, *evolution assumption*, *derivative-bound* part, and *derivative-free* part. We use them to describe what kinds of PDEs can be discovered with current methods and to motivate our new class of differential equations. (Section 3)
- We propose a new general class of PDEs (*Variational-Ready* PDEs) that admit the variational formulation (and thus allows to circumvent the derivative estimation). We also prove a theorem that motivates a novel objective function. (Section 5)
- We use the novel objective function to develop D-CIPHER, a new algorithm that searches over the *Variational-Ready* PDEs. (Section 6)

In addition to the main contributions above, we also develop a new optimization procedure (CoLLie) to help D-CIPHER search through this space efficiently. (Section 7)

## 2 Preliminaries

In this section, we provide background information about PDEs and their variational formulation.

**Notation and definitions.** We denote the set $\{1, 2, \ldots, n\}$ as $[n]$ and the set of non-negative integers as $\mathbb{N}_0$. Throughout this paper we let $M, N, K \in \mathbb{N}$ be some natural numbers and let $\Omega \subset \mathbb{R}^M$ be an open set inside $\mathbb{R}^M$. A comprehensive table with all symbols used in this work can be found in Appendix A together with some definitions restated more formally.

**Going beyond ODEs.** The simplest differential equations are ordinary differential equations that describe quantities that evolve with respect to only one independent variable, usually time. Most methods assume that the ODE is explicit and can be represented as a system of first-order ODEs:

$$\dot{u}_j(t) = f_j(t, \boldsymbol{u}(t)) \tag{1}$$

where $\dot{u}_j$ represents the derivative of $u_j$. Then the discovery problem is reduced to deciding the order of the derivative (usually first or second) and the discovery of $f_j$.

For PDEs, it is not enough to talk about *the* derivative, as we can take derivatives with respect to different variables. We denote the *mixed derivative* as $\partial^{\boldsymbol{\alpha}}$, where $\boldsymbol{\alpha} \in \mathbb{N}_0^M$ is called a multi-index, and define it as $\partial^{\boldsymbol{\alpha}} = \partial_1^{\alpha_1} \partial_2^{\alpha_2} \ldots \partial_M^{\alpha_M}$. Each $\partial_i^{\alpha_i} = \partial^{\alpha_i}/\partial x_i^{\alpha_i}$ is a $\alpha_i^{\text{th}}$-order partial derivative with respect to $x_i$ (the $i^{\text{th}}$ independent variable)[2]. We define the order of $\boldsymbol{\alpha}$ as $|\boldsymbol{\alpha}| = \sum_{i=1}^M \alpha_i$. We call $\partial^{\boldsymbol{\alpha}}$ *non-trivial* if $|\boldsymbol{\alpha}| > 0$.

---

[2]Throughout this work we assume that the functions we use are smooth enough for the *equality of mixed partials* [50] to hold. In that case, any mixed derivative can be uniquely specified by a multi-index.

A PDE of order $K$ is any equation of the form

$$f(\boldsymbol{x}, \boldsymbol{u}(\boldsymbol{x}), \partial^{[K]}\boldsymbol{u}(\boldsymbol{x})) = 0 \ \forall \boldsymbol{x} \in \Omega \tag{2}$$

where $\partial^{[K]}\boldsymbol{u}$ are all non-trivial mixed derivatives of all $u_j$ ($j \in [N]$) up to the $K^{\text{th}}$ order. We call a PDE *closed-form* if $f$ is closed-form.

**Variational formulation** (VF) of PDEs is a way to describe PDEs without referring to their derivatives [19]. It works as follows: we take a differential equation, we multiply it by a special *testing function*, and integrate. Finally, we perform integration by parts to move the derivatives from the dependent variable $u$ onto the testing functions. E.g., for a homogeneous heat equation,

$$\partial_t u - \theta \partial_x^2 u = 0 \iff \int_{\mathbb{R}^2} (\partial_t u - \theta \partial_x^2 u)\phi \, dt dx = 0 \ \forall \phi \iff \int_{\mathbb{R}^2} -u\partial_t \phi + \theta u \partial_x^2 \phi \, dt dx = 0 \ \forall \phi \tag{3}$$

For more details, see Appendix A and B. By not depending on the derivatives, methods that utilize VF are more robust to noise than their derivative-estimating counterparts [42, 35, 36].

# 3 Relaxing assumptions while staying robust to noise

Below, we introduce the *evolution assumption* (EA) and the *linear combination* (LC) assumption made by the current discovery methods.

**Evolution Assumption.** Although there is no generally accepted notion of an explicit PDE (as is the case for ODEs), we define an *evolution form* of a PDE to be an equation of the form

$$\partial^{\boldsymbol{\alpha}} u_j(\boldsymbol{x}) = f(\boldsymbol{x}, \boldsymbol{u}(\boldsymbol{x}), \partial^{[K]/\boldsymbol{\alpha}}\boldsymbol{u}(\boldsymbol{x})) \ \forall \boldsymbol{x} \in \Omega \tag{4}$$

where $\partial^{[K]/\boldsymbol{\alpha}}$ is $\partial^{[K]}$ with $\partial^{\boldsymbol{\alpha}}$ omitted, $\boldsymbol{\alpha}$ is a known multi-index and $j \in [N]$. Note that if $M = 1$ and $|\boldsymbol{\alpha}| = K$ then Equation 4 becomes exactly the definition of an explicit ODE.

In fact, many algorithms for PDE discovery assume a particular evolution form [35]. We call it an *evolution assumption* (EA). However, this assumption requires the knowledge of $\boldsymbol{\alpha}$ and $j$ which might not be trivial. Usually, $\partial^{\boldsymbol{\alpha}}$ is assumed to be the first derivative with respect to time ($\partial_t$) [47] but it is not the case for many well-known PDEs such as the wave equation or Gauss's law.

**Linear combinations.** Current PDE discovery algorithms [47, 35, 12] consider PDEs that are linear in parameters. That means the PDE can be represented as a linear combination of functions, i.e.,

$$\sum_{p=1}^{P} \theta_p f_p(\boldsymbol{x}, \boldsymbol{u}(\boldsymbol{x}), \partial^{[K]}\boldsymbol{u}(\boldsymbol{x})) = 0 \ \forall \boldsymbol{x} \in \Omega \tag{5}$$

where $\theta_p \in \mathbb{R}$ for $p \in [P]$ are the only constants that are optimized. As there are lot of expressions that cannot be put in that form, these algorithms fail to discover more complex equations. In particular, for an unknown $\theta \in \mathbb{R}$ functions such as $\sin(\theta x_i)$, $e^{\theta x_i}$ or $\frac{1}{x_i + \theta}$ cannot be learned by these algorithms.

**How to relax these assumptions and still allow for variational formulation?** Current methods that utilize VF either assume that the PDE is in an LC form or they only work for explicit first-order ODEs. Moreover, all of them also make the evolution assumption. Relaxing LC is not trivial because not all PDEs admit VF. As in Equation 3, the PDE has to be a sum of terms for which the integration by parts can be performed. Our crucial observation is that for any term that does not contain any derivatives (and thus does not need to be integrated by parts) *no additional constraints* need to be put in place. Due to the significance of these terms, we propose the following characterization of a PDE.

**Derivative-bound part and derivative-free part.** Any PDE can be expressed in the form

$$f(\boldsymbol{x}, \boldsymbol{u}(\boldsymbol{x}), \partial^{[K]}\boldsymbol{u}(\boldsymbol{x})) - g(\boldsymbol{x}, \boldsymbol{u}(\boldsymbol{x})) = 0 \ \forall \boldsymbol{x} \in \Omega \tag{6}$$

where we collect all the terms with the derivatives into $f(\boldsymbol{x}, \boldsymbol{u}(\boldsymbol{x}), \partial^{[K]}\boldsymbol{u}(\boldsymbol{x}))$ and all terms without the derivatives into $g(\boldsymbol{x}, \boldsymbol{u}(\boldsymbol{x}))$. We call $f$ the *derivative-bound* part and $g$ the *derivative-free* part (denoted also *∂-bound* and *∂-free*). ∂-free part can be evaluated directly given $\boldsymbol{u}$, whereas the ∂-bound part requires access to the derivatives. Note that for first-order ODEs, $f$ is trivial and equal to $\dot{u}_j$.

**Constraints on the derivative-bound part.** Although VF does not require any constraints on the ∂-free part, we still need to put some constraints on the ∂-bound part for the integration by parts to

Table 1: Columns correspond to challenges outlined in Section 1 and answer the following questions: Can it discover PDEs? Does it avoid derivative estimation? Is the evolution assumption unnecessary (Equation 4)? Can it discover any closed-form $\partial$-free part (Equation 6)?

| Method | PDEs | No $\partial$ estimation | No evolution assumption | Any closed-form $\partial$-free part |
|---|---|---|---|---|
| SINDy [10] | ✗ | ✗ | ✗ | ✗ |
| SINDy-implicit [33] | ✗ | ✗ | ✓ | ✗ |
| PDE-FIND [47] | ✓ | ✗ | ✗ | ✗ |
| PDE-Net 2.0 [29] | ✓ | ✗ | ✗ | ✗ |
| WSINDy [35, 46] | ✓ | ✓ | ✗ | ✗ |
| D-CODE [42] | ✗ | ✓ | ✗ | ✓ |
| D-CIPHER | ✓ | ✓ | ✓ | ✓ |

work. This is what we do in Section 5, where we aim to define currently the broadest form of PDEs that admit the variational formulation using the above characterization.

**Optimization challenge.** D-CIPHER does *not* need the evolution assumption and it can even discover some PDEs that cannot be put into the evolution form. Moreover, unlike previous methods, D-CIPHER is *not* limited to PDEs that can be represented as a linear combination of functions (we describe the exact form we assume in Section 6). This makes the optimization problem much harder as we search over *all* closed-form functions $g$ and for each candidate, we try to find the best counterpart $f$ among the allowed expressions. This is very different from the previous approaches, which either do not need to find $f$ as they work only for first-order ODEs [42] or they constrain equally both the $\partial$-bound part and $\partial$-free part to be a linear combination of some pre-specified functions [35] (for more details, see Table 10, and Table 11 in Appendix F). One way we address this challenge is by developing a new optimization procedure (Section 7).

## 4 Related works

**Symbolic Regression.** The goal of symbolic regression is to find a closed-form expression that best models the given dataset both in terms of accuracy and simplicity. In contrast with the conventional regression analysis which optimizes the parameters of a pre-specified model, symbolic regression aims at discovering both the general structure and the parameters of the model. Most of the existing work focuses on developing optimization algorithms. Genetic Programming [26] has been widely used for that task [49]. A different strategy has been employed in AI Feynman [54, 55] that uses neural networks to reduce the search space by identifying simplifying properties like symmetry or separability. Optimization methods based on pre-trained neural networks [4, 20], reinforcement learning [40], and Meijer-G functions [1, 13] have also been proposed.

**Data-driven discovery of closed-form differential equations.** Data-driven discovery of physical laws is an established area of machine learning [6, 49]. The pioneering work in that area was SINDy [10] that constrained the space of equations to linear combinations of functions from a predefined library and used sparse regression to discover explicit ODEs. It was later extended to include implicit ODEs [33, 23] and PDEs [47, 48]. Various other extensions were proposed by improving the derivative estimation and the training procedure [45, 61], adding additional selection criteria [32] and learning the library using genetic programming [34, 12, 62]. A different approach is taken by [29] (an extension of [30]) which uses convolutional and symbolic neural networks. It is important to note that *all of these methods still assume the PDE to be a linear combination* as discussed in Section 3 (Equation 5) which significantly limits their search space. Some other developments are based on Gaussian processes [44, 43] but they require the exact form of the PDE and only optimize the parameters.

**Variational approach.** Recently, the variational approach has been used as a viable alternative to derivative estimation. However, they have only been used for differential equations in a linear combination form [35, 36, 46] or closed-form first-order ODEs [42]. Extending the variational approach to closed-form PDEs is not trivial as PDEs are much more complex than ODEs and not all closed-form PDEs admit the variational formulation. In fact, the approaches that learn the library mentioned in the previous paragraph can produce exactly such terms which prohibits the use of variational formulation. To address these challenges we use the new notions defined in Section 3 to define a new and general class of PDEs in Section 5 that admit the variational formulation.

# 5 Variational-Ready PDEs

In this section, we propose a new and very general class of PDEs, the *Variational-Ready* PDEs (VR-PDEs), which can be characterized *without* referring to the derivative. The VR-PDEs allow arbitrary $\partial$-free part but make some minor restrictions on the $\partial$-bound part. These restrictions allow one to use the variational formulation of PDEs to circumvent derivative estimation entirely. Despite the minor restriction, VR-PDEs contain many well-known PDEs, including all linear PDEs, Maxwell's equations, and Navier-Stokes equations (additional examples provided in Appendix B).

To define the new class of PDEs, we need the following definition.

**Definition 1** (Extended derivative and differential operator). Let $\boldsymbol{\alpha} \in \mathbb{N}_0^M$, $|\boldsymbol{\alpha}| \leq K$, be a multi-index. Let $h : \mathbb{R}^{M+N} \to \mathbb{R}$ and $a : \mathbb{R}^M \to \mathbb{R}$ be smooth functions. An *extended derivative* $\mathcal{E}$, denoted $(\boldsymbol{\alpha}, a, h)$, maps a vector field $\boldsymbol{u} : \mathbb{R}^M \to \mathbb{R}^N$ to a function $\mathcal{E}[\boldsymbol{u}] : \mathbb{R}^M \to \mathbb{R}$ defined as:

$$\mathcal{E}[\boldsymbol{u}](\boldsymbol{x}) = a(\boldsymbol{x})\partial^{\boldsymbol{\alpha}}[h(\boldsymbol{x}, \boldsymbol{u}(\boldsymbol{x}))] \tag{7}$$

$\mathcal{E}$ is called *closed-form* if $a$ and $h$ are closed-form. We call $\mathcal{E}$ *non-degenerate* if $|\boldsymbol{\alpha}| > 0$.

Now, let $(\mathcal{E}_p)_{p \in [P]}$ be a finite sequence of non-degenerate extended derivatives. The extended differential operator, denoted as $\mathcal{E}_{[P]}$ is an operator defined as:

$$\mathcal{E}_{[P]}[\boldsymbol{u}](\boldsymbol{x}) = \sum_{p=1}^{P} \mathcal{E}_p[\boldsymbol{u}](\boldsymbol{x}) \tag{8}$$

*Remark.* Any linear operator $L = \sum_{\boldsymbol{\alpha} \in \mathcal{A}} a_{\boldsymbol{\alpha}} \partial^{\boldsymbol{\alpha}}$ acting on $u_j$ is an extended differential operator.

**Definition 2** (Variational-Ready PDE). Let $\mathcal{E}_{[P]}$ be an extended differential operator, and let $g : \mathbb{R}^{M+N} \to \mathbb{R}$ be a continuous function. We denote a *Variational-Ready* PDE (VR-PDE) by a pair $(\mathcal{E}_{[P]}, g)$ and define it as:

$$\mathcal{E}_{[P]}[\boldsymbol{u}](\boldsymbol{x}) - g(\boldsymbol{x}, \boldsymbol{u}(\boldsymbol{x})) = 0 \ \forall \boldsymbol{x} \in \Omega \tag{9}$$

We extend the standard variational formulation of PDEs (Proposition 1 in Appendix B) from linear PDEs to all VR-PDEs. The following definition is useful in further discussion.

**Definition 3.** Consider a field $\boldsymbol{u} : \Omega \to \mathbb{R}^N$, and an extended derivative $\mathcal{E} = (\boldsymbol{\alpha}, a, h)$. Let $\phi : \Omega \to \mathbb{R}$ be a *testing function* ($\mathcal{C}^K$ function [3] with compact support). We define the functional

$$\mathcal{F}(\mathcal{E}, \boldsymbol{u}, \phi) = \int_{\Omega} h(\boldsymbol{x}, \boldsymbol{u}(\boldsymbol{x}))(-1)^{|\boldsymbol{\alpha}|}\partial^{\boldsymbol{\alpha}}[a(\boldsymbol{x})\phi(\boldsymbol{x})]d\boldsymbol{x}$$

We can now use this functional to formulate variational characterization of VR-PDEs.

**Theorem 1.** $\boldsymbol{u} : \Omega \to \mathbb{R}^N$, where $u_j \in \mathcal{C}^K$, is a solution to a VR-PDE in Equation 9 if and only if

$$\sum_{p=1}^{P} \mathcal{F}(\mathcal{E}_p, \boldsymbol{u}, \phi) - \int_{\Omega} [g(\boldsymbol{x}, \boldsymbol{u}(\boldsymbol{x}))\phi(\boldsymbol{x})] \, d\boldsymbol{x} = 0 \tag{10}$$

*for all testing functions $\phi : \Omega \to \mathbb{R}$.*

*Proof.* Appendix B. $\square$

This theorem motivates the *variational loss function* as we expect the left-hand side of Equation 10 to be closer to 0 the closer the candidate PDE is to the true one. To calculate how well a set of vector fields $\mathcal{D} = \{\boldsymbol{u}^{(d)}\}_{d=1}^{D}$ matches a VR-PDE $(\mathcal{E}_{[P]}, g)$ we propose the following loss function.

$$\mathcal{L}\left(\mathcal{E}_{[P]}, g\right) = \sum_{d=1}^{D} \sum_{s=1}^{S} \left( \sum_{p=1}^{P} \mathcal{F}(\mathcal{E}_p, \boldsymbol{u}^{(d)}, \phi_s) - \int_{\Omega} g(\boldsymbol{x}, \boldsymbol{u}^{(d)}(\boldsymbol{x}))\phi_s(\boldsymbol{x})d\boldsymbol{x} \right)^2 \tag{11}$$

where $\{\phi_s\}_{s=1}^{S}$ is a set of predefined testing functions.

This novel loss function makes it possible to evaluate to what extent any VR-PDE matches the observed data. This loss can be used as an optimization objective in any algorithm that searches over some subspace of closed-form VR-PDEs. We propose D-CIPHER in Section 6 as an example of such an algorithm.

---

[3] We say $u : \mathbb{R}^M \to \mathbb{R}$ is in $\mathcal{C}^K$ if $\partial^{\boldsymbol{\alpha}} u$ exists and is continuous for all $|\boldsymbol{\alpha}| \leq K$.

# 6 D-CIPHER

In this section, we formulate the problem of PDE discovery and then we introduce a novel algorithm (D-CIPHER) to solve it. The diagram and pseudocode are presented in Figure 1 and in Appendix C.

**Problem formulation** We are given a dataset of *observed fields* $\mathcal{D} = \{\boldsymbol{v}^{(d)}\}_{d=1}^D$ with a finite *sampling grid* $\mathcal{G} \subset \Omega$. Each $\boldsymbol{v}^{(d)}(\boldsymbol{x})$ is a noisy measurement, i.e., $\boldsymbol{v}^{(d)} : \mathcal{G} \to \mathbb{R}^N$ is defined as

$$v_j^{(d)}(\boldsymbol{x}) = u_j^{(d)}(\boldsymbol{x}) + \epsilon_j^{(d)}(\boldsymbol{x}) \; \forall \boldsymbol{x} \in \mathcal{G} \; \forall j \in [N] \tag{12}$$

where $\epsilon_j^{(d)}(\boldsymbol{x})$ is a realization of a zero-mean random variable (noise), each $u_j^{(d)} : \Omega \to \mathbb{R}$ is a $\mathcal{C}^K$ function, and every *true field* $\boldsymbol{u}^{(d)}$ is governed by the same closed-form PDE $f$. The task is to infer the closed-form PDE, $f$, from the dataset $\mathcal{D} = \{\boldsymbol{v}^{(d)}\}_{d=1}^D$ and the sampling grid $\mathcal{G}$. We assume that $f$ is inside the class of closed-form VR-PDEs (Section 5) and its $\partial$-bound part is inside a subspace of extended differential operators spanned by a user-specified dictionary (see Step 1 below).

We propose an algorithm that consists of three steps. In the first step, we define the subspace of closed-form VR-PDEs we want to search over to reflect our knowledge of the problem. In the second step, we reconstruct the fields from noisy measurements. In the last step, we solve an optimization problem using a modified symbolic regression algorithm. For more details, check Appendix C.

**Step 1: Choose the form and incorporate prior knowledge.** A human expert should encode their prior knowledge of the problem into a dictionary of non-degenerate extended derivatives $\mathcal{Q}$ $= \{\hat{\mathcal{E}}_p\}_{p\in[P]}$. We use this dictionary to search over a finite-dimensional subspace of closed-form operators spanned by this set. In other words, we assume that the VR-PDE is of the form:

$$\sum_{p=1}^P \beta_p \hat{\mathcal{E}}_p[\boldsymbol{u}](\boldsymbol{x}) - g(\boldsymbol{x}, \boldsymbol{u}(\boldsymbol{x})) = 0 \; \forall \boldsymbol{x} \in \Omega \tag{13}$$

where $\boldsymbol{\beta} \in \mathbb{R}^P$, $g$ is *any* closed-form function of $M + N$ variables, and $\hat{\mathcal{E}}_p = (\boldsymbol{\alpha}_p, a_p, h_p)$.

For instance, a dictionary might include only the partial derivatives up to a certain order. For a 1+1 second-order equation that means $\mathcal{Q} = \{\partial_t, \partial_x, \partial_{tx}, \partial_t^2, \partial_x^2\}$. That is already enough to discover heat and wave equations with any closed-form source. If, for instance, the user suspects the presence of the advection term $uu_x$ (as in the Burgers' equation), the term $\partial_x(u^2)$ can be included in the library.

It's important to note that we do *not* assume any particular form of $g$ apart from being closed-form.

**Step 2: Estimate the fields.** As the dataset $\mathcal{D}$ consists of noisy and infrequently sampled fields, we first need to estimate the "true" fields $\hat{\boldsymbol{u}}^{(d)}$ from $\boldsymbol{v}^{(d)}$. Any choice of reconstruction algorithm can be used and the user should choose it according to the problem setting and their domain knowledge.

**Step 3: Optimize.** We minimize the loss function in Equation 11 for the estimated fields $\{\hat{\boldsymbol{u}}^{(d)}\}_{d=1}^D$ among all PDEs of the form in Equation 13. We solve the following optimization problem:

$$\min_g \min_{||\boldsymbol{\beta}||_1=1} \sum_{d=1}^D \sum_{s=1}^S \left( \sum_{p=1}^P \mathcal{F}(\beta_p \hat{\mathcal{E}}_p, \hat{\boldsymbol{u}}^{(d)}, \phi_s) - \int_\Omega g(\boldsymbol{x}, \hat{\boldsymbol{u}}^{(d)}(\boldsymbol{x}))\phi_s(\boldsymbol{x})d\boldsymbol{x} \right)^2 \tag{14}$$

As we want to discover both $g$ and $\boldsymbol{\beta}$ we cannot use the standard penalties on $\boldsymbol{\beta}$ such as the $\lambda||\boldsymbol{\beta}||_2$ or $\lambda||\boldsymbol{\beta}||_1$, as the loss would be minimized by $g = 0$ and $\boldsymbol{\beta} = \boldsymbol{0}$. Therefore we put the constraint $||\boldsymbol{\beta}||_1 = 1$. We choose the L1 norm to encourage sparsity in the coordinates of the vector $\boldsymbol{\beta}$.

The inner minimization in Equation 14 can be rewritten as a constrained least-squares problem.

$$\min_{||\boldsymbol{\beta}||_1=1} \sum_{(d,s)\in[D]\times[S]} \left( \boldsymbol{\beta} \cdot \boldsymbol{z}^{(d,s)} - w^{(d,s)} \right)^2 \tag{15}$$

where $\hat{\mathcal{E}}_p = (\boldsymbol{\alpha}_p, a_p, h_p)$ and $\boldsymbol{z}^{(d,s)} \in \mathbb{R}^P$, $w^{(d,s)} \in \mathbb{R}$ are defined as

$$z_p^{(d,s)} = \int_\Omega h_p(\boldsymbol{x}, \hat{\boldsymbol{u}}^{(d)}(\boldsymbol{x}))(-1)^{|\boldsymbol{\alpha}_p|}\partial^{\boldsymbol{\alpha}_p}(a_p(\boldsymbol{x})\phi_s(\boldsymbol{x}))d\boldsymbol{x}$$
$$w^{(d,s)} = \int_\Omega g(\boldsymbol{x}, \hat{\boldsymbol{u}}^{(d)}(\boldsymbol{x}))\phi_s(\boldsymbol{x})d\boldsymbol{x} \tag{16}$$

We show the full derivation in Appendix C. $z^{(d,s)}$ can be precomputed at the beginning of the algorithm without estimating the derivatives of the reconstructed fields. They can be easily calculated if the derivatives of the testing functions $\phi_s$ and the derivatives of $a_p$ can be analytically computed.

As the optimization problem in Equation 15 has to be solved many times for different closed-form expressions $g$, it poses some unique challenges. As standard approaches are not sufficiently fast, we design a new heuristic algorithm to solve this problem, CoLLie, and describe it in the next section.

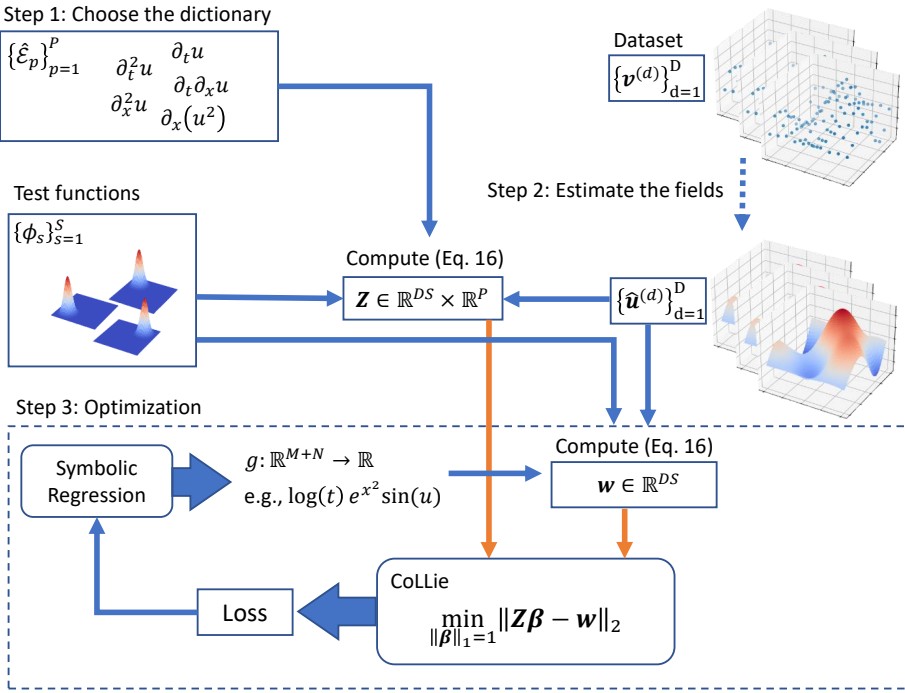

Figure 1: This diagram describes how the algorithm works. After the optimization procedure is finished, we get the best found closed-form function $g$ and use CoLLie to find the best vector $\boldsymbol{\beta}$. The found equation has the form $\sum_{p=1}^{P} \beta_p \hat{\mathcal{E}}_p[\boldsymbol{u}](\boldsymbol{x}) - g(\boldsymbol{x}, \boldsymbol{u}(\boldsymbol{x})) = 0$

## 7   CoLLie

The problem in Equation 15 from the previous section can be formulated as follows. Given matrix $\boldsymbol{A} \in \mathbb{R}^{m \times n}$ and vector $\boldsymbol{b} \in \mathbb{R}^m$, find a vector $\boldsymbol{z} \in \mathbb{R}^n$ that minimizes $||\boldsymbol{Az} - \boldsymbol{b}||_2^2$ such that $||\boldsymbol{z}||_1 = 1$. The task is challenging as the unit L1 sphere is not convex. A method that guarantees an optimal solution is based on an observation that the $(n-1)$-dimensional L1 sphere consists of $2^n$ $(n-1)$-simplices (which are convex). Minimizing $||\boldsymbol{Az} - \boldsymbol{b}||_2^2$ on a simplex is a quadratic program [8] with many available solvers [2, 51, 3]. However, that means that the computation time scales *exponentially* with the number of dimensions. This is prohibitively long for the inner optimization of our algorithm. Therefore, we design a heuristic algorithm CoLLie (**Co**nstrained **L**1 Norm **Lea**st Squares) that finds an approximate solution but is significantly faster (Figure 2). We observe that this optimization problem is related to the one encountered in LASSO. Denote $\boldsymbol{z_0}$ the solution that minimizes $||\boldsymbol{Az} - \boldsymbol{b}||_2^2$ (no constraints). If $||\boldsymbol{z_0}|| \geq 1$, the problem is equivalent to finding $\lambda$ (in the Lagrangian form of LASSO, Equation 47) such that the LASSO solution has the norm 1. Least Angle Regression (LARS) [17] is a popular algorithm used to minimize the LASSO objective that computes complete solution paths. These paths show how the coefficients of the solution change as $\lambda$ moves from 0 to $\lambda_{max}$ (from no constraints to effectively imposing the solution to be **0**). See Figure 6 in Appendix D.2. CoLLie uses these solution paths to calculate the exact solution to the optimization problem. The case $0 < ||\boldsymbol{z_0}|| < 1$ is harder as it corresponds to $\lambda < 0$. CoLLie addresses this challenge by extending the solution paths generated by LARS beyond $\lambda = 0$ for $\lambda < 0$. We assume that the paths continue to be piecewise linear and keep their slope (Figure 6 in Appendix

D.2). CoLLie then uses these assumptions to efficiently find an approximate solution. We provide a detailed description of CoLLie in Appendix D.

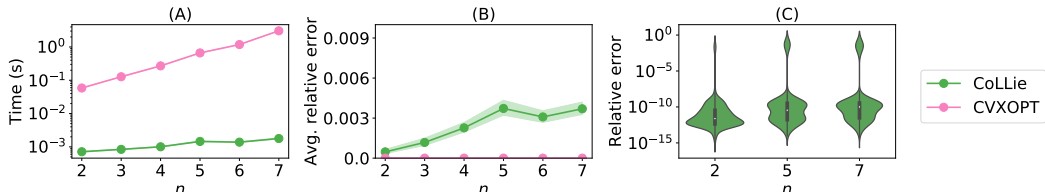

Figure 2: We compare CoLLie with an algorithm that uses CVXOPT [2] to solve each of the convex subproblems. We report the relative error between the loss obtained by CoLLie and the minimum loss achieved by CVXOPT. Panels **B** and **C** show the averages and the distributions of relative errors. The average relative error is below 0.005 and the bulk of the distribution is below $10^{-7}$. At the same time CoLLie is orders of magnitude faster (Panel **A**).

.

## 8 Experiments

We perform a series of synthetic experiments to show how well D-CIPHER is able to discover some well-known differential equations[4] (Table 2). First, we demonstrate that D-CIPHER performs better than current methods when discovering PDEs in a linear combination form (Section 8.1). Then we demonstrate it can discover PDEs with a closed-form $\partial$-free part that *cannot* be expressed as a linear combination and thus are beyond the capabilities of current methods (Section 8.2). We contrast D-CIPHER with its ablated version where the derivatives are estimated and the standard MSE loss is used instead of the variational loss (details in Appendix E.1). For additional information about the experiments (e.g., implementation details, data generation, experimental settings) see Appendix E.

Table 2: Equations used in the experiments. "LC" column specifies if the equation can be represented as a linear combination (Equation 5). "VR" column specifies if the PDE is Variational-Ready

| Name | Equation | LC | VR |
|---|---|---|---|
| Homogeneous heat equation | $\partial_t u - \theta_1 \partial_x^2 u = 0$ | ✓ | ✓ |
| Burger's equation | $\partial_t u + u\partial_x u - \theta_1 \partial_x^2 u = 0$ | ✓ | ✓ |
| Kuramoto-Sivashinsky equation | $\partial_t u + \partial_x^2 u + \partial_x^4 u + u\partial_x u = 0$ | ✓ | ✓ |
| Forced and damped harmonic oscillator | $\partial_t^2 + 2\theta_1\theta_2\partial_t u + \theta_2^2 u = \theta_3 \sin(\theta_4 t)$ | ✗ | ✓ |
| SLM model (Appendix F.1) | $\partial_t u + \partial_x u = -2e^{\theta_1 x}u$ | ✗ | ✓ |
| Inhomogeneous heat equation | $\partial_t u - \theta_1 \partial_x^2 u = \theta_2 e^{\theta_3 t}$ | ✗ | ✓ |
| Inhomogeneous wave equation | $\partial_t^2 u - \theta_1 \partial_x^2 u = \theta_2 e^t \sin(\theta_3 t)$ | ✗ | ✓ |

**Evaluation metrics.** To establish how well a discovered PDE matches the ground truth, we evaluate its $\partial$-free and $\partial$-bound parts separately. For the $\partial$-free part, we assign a binary variable indicating whether the correct functional form of the equation was recovered (please check Appendix E.8 for details). For the $\partial$-bound part, we measure the RMSE between the found coefficients of $\beta$ and the target ones. We report the averages and standard deviations for both parts. We call the averages respectively **Success Probability** and **Average RMSE**.

**Implementation.** We use B-Splines [15] as the testing functions and we estimate the fields in Step 2 of D-CIPHER with a Gaussian Process [60]. The outer optimization in Step 3 is performed using a modified genetic programming algorithm [26] and the inner optimization by CoLLie (Section 7). We also show additional experiments with different estimation algorithms in Appendix F.

### 8.1 Discovering Linear Combinations: comparison with other methods

We compare D-CIPHER against two variants of PDE-FIND [47] and WSINDy [46] (as implemented in PySINDy library [24, 16]) with optimization performed by Stepwise Sparse Regression [7] or

---

[4]All experiment code can be found at `https://github.com/krzysztof-kacprzyk/D-CIPHER`

Forward Regression Orthogonal Least-Squares [5]. We note that D-CIPHER is specifically designed to discover PDEs that are beyond the capabilities of current methods, i.e., where the derivative-free part can be any closed-form expression. Current methods are usually tested on equations where the derivative-free part is trivial (identically equal to 0). Even though these algorithms are specialized to discover these simpler kinds of equations, D-CIPHER performs better than (or equally well as) PDE-FIND and WSINDy, regardless of the optimization algorithm, when tested on Burgers' equation the homogeneous heat equation, and Kuramoto–Sivashinsky equation (Figure 3). This demonstrates gain from both the variational loss and the new optimization routine.

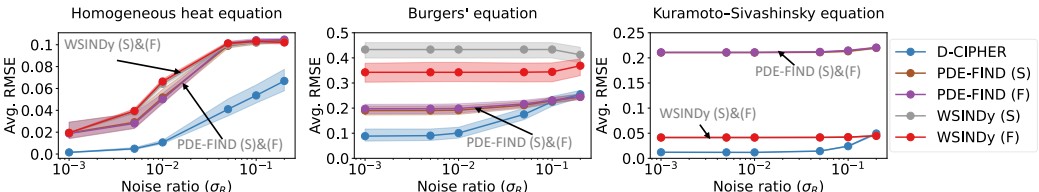

Figure 3: Simulation results for the Burgers' equation, homogeneous heat equation, and Kuramoto–Sivashinsky equation. We report the Average RMSE of the $\partial$-bound part of the equation. Note that some of the benchmarks overlap

.

## 8.2 Discovering equations beyond current methods

**Forced and damped harmonic oscillator.** As the oscillator is described by a second-order ODE, it cannot be discovered by D-CODE [42]. D-CIPHER discovers the correct functional form of the $\partial$-free part and achieves a low RMSE for the coefficients of $\boldsymbol{\beta}$ in most of the experimental settings. The performance is higher than or comparable to the ablated version of D-CIPHER, thus demonstrating gain from using the variational approach. We present the results in Figure 4.

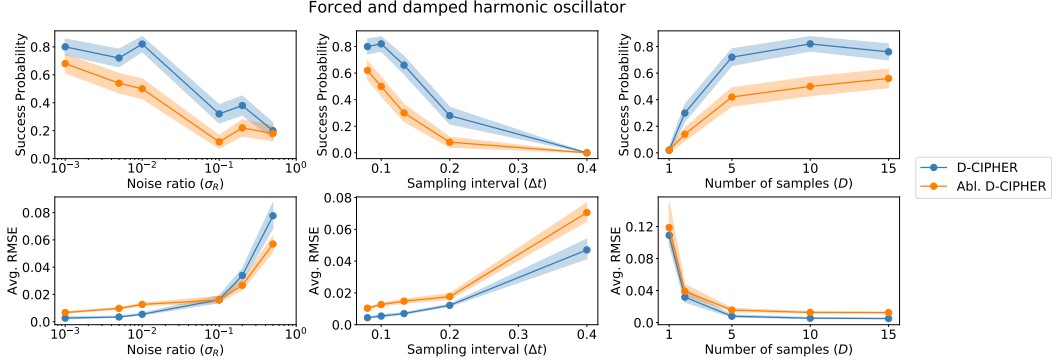

Figure 4: Success probability of discovering the correct $\partial$-free part of the equation and the average RMSE between the recovered $\partial$-bound part and the target one across different experimental settings. We compare D-CIPHER against its ablated version (Abl. D-CIPHER).

**Inhomogeneous heat equation.** D-CIPHER is able to discover the correct equation even in settings with very high noise. It performs better than the ablated version, thus showing the importance of the variational objective. The result are presented in Table 3.

Table 3: We report the success probability of discovering the $\partial$-free part and the Average RMSE of the $\partial$-bound part for the inhomogeneous heat equation. Standard deviations shown in brackets.

| Method | Success probability | | | Average RMSE | | |
| --- | --- | --- | --- | --- | --- | --- |
| | $\sigma_R = 0.05$ | 0.1 | 0.2 | $\sigma_R = 0.05$ | 0.1 | 0.2 |
| D-CIPHER | 0.64 (.07) | 0.42 (.07) | 0.12 (.05) | 0.15 (.009) | 0.21 (.007) | 0.24 (.005) |
| Ablated D-CIPHER | 0.46 (.07) | 0.20 (.06) | 0.04 (.03) | 0.18 (.009) | 0.24 (.008) | 0.27 (.007) |

**Inhomogeneous wave equation.** This equation does not have the standard evolution form, as it does not involve the $\partial_t$ term. Thus, even without the source term, most of the current methods cannot be applied directly to discover this equation. In Figure 5 we show the absolute difference between the true field and the fields computed from the sources discovered by D-CIPHER and its ablated version across different measurement settings. D-CIPHER finds the correct functional form with coefficients not far from the ground truth. The ablated version fails to discover the correct functional form and the found $\partial$-free part does not reproduce the correct behavior of the equation.

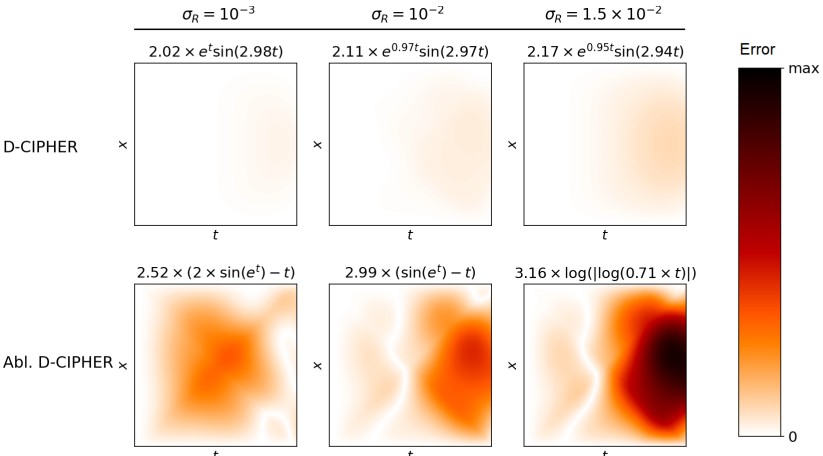

Figure 5: We solve the inhomogeneous wave equation for the $\partial$-free parts found by the D-CIPHER and its ablated version Abl. D-CIPHER. We show the absolute difference between the computed fields and the true field generated by $\partial$-free part $2 \times e^t \sin(3t)$.

## 9 Discussion

**Applications.** As D-CIPHER can potentially discover any closed-form $\partial$-free part, it is especially useful when this part of the PDE captures an essential component of the phenomenon. We demonstrate it by finding the heat and vibration sources as well as the driving force of an oscillator. Beyond the spatio-temporal physical equations, D-CIPHER might prove useful in discovering population models structured by age, size, and spatial position [59, 58], age-dependent epidemiological models [21], and predator-prey models with age-structure [41]. All these equations are VR-PDEs where the $\partial$-free parts are crucial elements of the equations signifying the rates of mortality, infection, recovery, or growth. We believe that discovering closed-form equations for these systems would prove invaluable in understanding their behavior.

**Limitations and open challenges.** D-CIPHER may fail in some scenarios, either due to *challenging experimental settings* or a *challenging underlying PDE*. Challenging experimental settings might include unobserved variables, high measurement noise, infrequent sampling, and inadequate domain (e.g., small time horizon). Challenging PDE forms might include a PDE outside of the VR-PDE class or a $\partial$-free part with a complex expression that is difficult to find. We note that we address some of these challenges by utilizing a variational approach, defining VR-PDEs to be a very general class of equations, and designing CoLLie, enabling a thorough search across closed-form expressions.

**Ethics Statement.** We want to emphasize that D-CIPHER was designed to facilitate the process of scientific discovery by extracting closed-form PDEs from data. It is not intended to or capable of replacing human experts in the modeling process. No human-derived data was used.

## Acknowledgments and disclosure of funding

This work is supported by Cancer Research UK and Roche. We want to thank Samuel Holt, Jonathan Crabbé, and anonymous reviewers for their useful comments and feedback on earlier versions of this work. We are grateful to Professor Yuanzhang Xiao for insightful discussions about the optimization algorithms.

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

## Table of supplementary materials

## A  Notation and definitions

### A.1  Notation

Table 4: Symbols used in this work

| Symbol | Meaning |
|--------|---------|
| $[n]$ | a set of numbers $\{1, \ldots, n\}$ |
| $\mathbb{N}$ | a set of natural numbers, i.e., $\{1, 2, 3, \ldots, \}$ |
| $\mathbb{N}_0$ | a set of non-negative integers, i.e., $\{0, 1, 2, 3, \ldots\}$ |
| $M$ | the dimension of the domain of a vector field |
| $N$ | the dimension of the codomain of a vector field |
| $K$ | denotes the smoothness of functions or the maximum order of derivatives |
| $D$ | the size of the dataset of observed fields |
| $S$ | the number of testing functions |
| $\Omega$ | an open set in $\mathbb{R}^M$ |
| $\dot{u}(t)$ | the derivative of $u$ at $t$ |
| $\boldsymbol{\alpha}$ | a multi-index, an element of $\mathbb{N}_0^M$ |
| $|\boldsymbol{\alpha}|$ | the order of $\boldsymbol{\alpha}$, $|\boldsymbol{\alpha}| = \sum_i^M \alpha_i$ |
| $\partial_i^{\alpha_i}$ | $\alpha_i^{\text{th}}$-order partial derivative with respect to the $i^{\text{th}}$ variable |
| $\partial^{\boldsymbol{\alpha}}$ | $\partial_1^{\alpha_1} \partial_2^{\alpha_2} \ldots \partial_M^{\alpha_M}$ |
| $\mathcal{C}^K$ | a set of functions with continuous partial derivatives $\partial^{\boldsymbol{\alpha}}$ for all $|\boldsymbol{\alpha}| \leq K$ |
| $\mathcal{E}$ | an extended derivative, Definition 1 |
| $\mathcal{E}_{[P]}$ | an extended differential operator, Definition 1 |
| $\mathcal{F}$ | the functional used in the variational loss, Definition 3 |
| $\mathcal{L}(\mathcal{E}_{[P]}, g)$ | the variational loss, Equation 11 |
| $\mathcal{G}$ | a sampling grid, Definition 10 |
| $\boldsymbol{u}$ | a true field, Definition 9 |
| $\boldsymbol{v}$ | an observed field, Definition 10 |
| $\mathcal{D}$ | a dataset of observed trajectories |
| $\epsilon$ | the noise |
| $\mathcal{Q}$ | a dictionary of non-degenerate extended derivatives |
| $\boldsymbol{\beta}$ | a vector describing the $\partial$-bound part of the VR-PDE |
| $\sigma_R$ | a noise ratio |
| $\Delta t$ | a sampling interval |

### A.2  Definitions

In this section, we collect the definitions of some of the important terms used in the paper for easy reference.

**Definition 4** (Closed-form expressions and functions). A *closed-form* expression is a mathematical expression that consists of a finite number of variables, constants, arithmetic operations, and certain well-known functions (e.g., logarithm, trigonometric functions). A function $f$ is called *closed-form* if it can be represented by a closed-form expression. E.g., $f(x, y) = x^2 \log(y) + \sin(3z)$.

*Remark.* In practice, we do not want to consider any finite expression. Any symbolic regression algorithm penalizes expressions that are too long putting a soft constraint on the number of elements used. That is why deep neural networks are not considered closed-form even if they satisfy the conditions in Definition 4.

**Definition 5** (Multi-index). An $n$-dimensional *multi-index* $\boldsymbol{\alpha}$ is an $n$-tuple

$$\boldsymbol{\alpha} = (\alpha_1, \alpha_2, \ldots, \alpha_n)$$

where $\forall i \in [n]\ \alpha_i \in \mathbb{N}_0$. Thus $\boldsymbol{\alpha} \in \mathbb{N}_0^n$. We define the order of $\boldsymbol{\alpha}$ as $|\boldsymbol{\alpha}| = \sum_{i=1}^n \alpha_i$.

**Definition 6.** For any $n$-dimensional multi-index $\boldsymbol{\alpha}$ we define a mixed derivative

$$\partial^{\boldsymbol{\alpha}} = \partial_1^{\alpha_1} \partial_2^{\alpha_2} \ldots \partial_n^{\alpha_n}$$

where $\partial_i^{\alpha_i} = \partial^{\alpha_i}/\partial x_i^{\alpha_i}$ is a $\alpha_i^{\text{th}}$-order partial derivative with respect to $x_i$ (the $i^{\text{th}}$ independent variable). We call $\partial^{\boldsymbol{\alpha}}$ *non-trivial* if $|\boldsymbol{\alpha}| > 0$. We denote the list of all non-trivial partial derivatives of $u$ up to order $K$ as $\partial^{[K]}u$.

**Definition 7** (Closed-form Partial Differential Equation). Let $f$ be a closed-form real smooth function. We say that a *vector field* $\boldsymbol{u} : \Omega \to \mathbb{R}^N$ is *governed* by a $K^{\text{th}}$-order closed-form PDE described by $f$ if

$$f(\boldsymbol{x}, \boldsymbol{u}(\boldsymbol{x}), \partial^{[K]}\boldsymbol{u}(\boldsymbol{x})) = 0\ \forall \boldsymbol{x} \in \Omega \tag{17}$$

where $\partial^{[K]}\boldsymbol{u}$ are all non-trivial mixed derivatives of all $u_j$ ($j \in [N]$) up to the $K^{\text{th}}$ order.

**Definition 8** (Testing function). Support of a function $\phi : \Omega \to \mathbb{R}$ is defined as

$$supp\ \phi = \overline{\{\boldsymbol{x} \in \Omega : \phi(\boldsymbol{x}) \neq 0\}}$$

where $\overline{\mathcal{B}}$ is the topological *closure* of $\mathcal{B}$ in $\Omega$.

$\phi$ is called a *testing function* if it is a $\mathcal{C}^K$ function with *compact* support.

**Definition 9** (True Field). We define a *true field* on $\Omega$ as a vector valued function $\boldsymbol{u} : \Omega \to \mathbb{R}^N$ where each $u_j : \Omega \to \mathbb{R}$ is a $\mathcal{C}^K$ function.

**Definition 10** (Observed field and sampling grid). We define a *sampling grid* $\mathcal{G}$ to be a finite subset of $\Omega$. Let $\boldsymbol{u} : \Omega \to \mathbb{R}^N$ be a true field on $\Omega$. An *observed field sampled from* $\boldsymbol{u}$ on a grid $\mathcal{G}$ is a function $\boldsymbol{v} : \mathcal{G} \to \mathbb{R}^N$ of the form

$$v_j(\boldsymbol{x}) = u_j(\boldsymbol{x}) + \epsilon_j(\boldsymbol{x})\ \forall \boldsymbol{x} \in \mathcal{G}\ \forall j \in [N]$$

where $\epsilon_j(\boldsymbol{x})$ corresponds to *noise*, a realisation of a zero-mean random variable.

**Definition 11** ($L1$ sphere). Let $n \in \mathbb{N}$. We define $n$-dimensional $L1$ sphere to be a subset of $\mathbb{R}^{n+1}$ defined as:

$$\{\boldsymbol{x} \in \mathbb{R}^{n+1} \mid ||x||_1 = 1\} \subset \mathbb{R}^{n+1} \tag{18}$$

**Definition 12** (Standard simplex). Let $n \in \mathbb{N}$. We define standard $n$-simplex to be a subset of $\mathbb{R}^{n+1}$ defined as:

$$\{\boldsymbol{x} \in \mathbb{R}^{n+1} \mid \sum_{i=1}^{n+1} x_i = 1 \wedge x_i \geq 0\ \forall i \in [n+1]\} \subset \mathbb{R}^{n+1} \tag{19}$$

# B  Variational-Ready PDEs

## B.1  Variational Formulation of PDEs

In this section, we provide the standard variational formulation of PDEs for linear PDEs [19].

**Definition 13** (Linear differential operator). Let $\mathcal{A}$ be a finite set of multi-indices. A linear differential operator $L$ is defined as

$$L = \sum_{\boldsymbol{\alpha} \in \mathcal{A}} a_{\boldsymbol{\alpha}} \partial^{\boldsymbol{\alpha}}$$

where $a_{\boldsymbol{\alpha}} \in \mathcal{C}^K$ is a non-zero sufficiently smooth function of dependent variables. If $\max_{\boldsymbol{\alpha} \in \mathcal{A}} |\boldsymbol{\alpha}| = n$ then we call $L$ an $n^{\text{th}}$-order linear differential operator. If all $a_{\boldsymbol{\alpha}}$ are constants we say that $L$ has *constant coefficients*.

The *adjoint* of $L$, denoted $L^\dagger$, is a linear differential operator defined as

$$L^\dagger u(\boldsymbol{x}) = \sum_{\boldsymbol{\alpha} \in \mathcal{A}} (-1)^{|\boldsymbol{\alpha}|} \partial^{\boldsymbol{\alpha}}(a_{\boldsymbol{\alpha}}(\boldsymbol{x})u(\boldsymbol{x})) \tag{20}$$

**Proposition 1** (Variational Formulation of PDEs for linear PDEs). *Let $K \in \mathbb{N}$. Consider a scalar field $u : \Omega \to \mathbb{R}$, such that $u \in \mathcal{C}^K$, a $K^{th}$-order linear differential operator $L$, and a continuous function $g : \Omega \to \mathbb{R}$. Let $\phi : \Omega \to \mathbb{R}$ be a testing function. Then $u$ satisfies a linear PDE*

$$L[u(\boldsymbol{x})] - g(\boldsymbol{x}) = 0 \;\forall \boldsymbol{x} \in \Omega \tag{21}$$

*if and only if*

$$\int_\Omega \left[ u(\boldsymbol{x}) L^\dagger \phi(\boldsymbol{x}) - g(\boldsymbol{x})\phi(\boldsymbol{x}) \right] d\boldsymbol{x} = 0 \tag{22}$$

*for all testing functions $\phi : \Omega \to \mathbb{R}$.*

*Note that the integrals are always well-defined as $\phi$ has a compact support.*

## B.2   Theorem

Before we prove the Theorem 1, we need the following lemma, which is a particular formulation of the Fundamental lemma of calculus of variations [18]. We also need a generalized version of the divergence theorem [31].

**Lemma** (Fundamental lemma of calculus of variations). *Let $K \in \mathbb{N}$, $\Omega$ be an open set in $\mathbb{R}^M$, and $u : \Omega \to \mathbb{R}$ be a continuous function. Then $u$ is equal to $0$ on the whole $\Omega$ if and only if $\int_\Omega u(\boldsymbol{x})\phi(\boldsymbol{x})d\boldsymbol{x} = 0$ for all $\mathcal{C}^K$ functions $\phi : \Omega \to \mathbb{R}$ with compact support.*

*Proof.* If $u$ is identically $0$ on $\Omega$ then all the integrals are trivially equal to $0$.

We now prove the converse.

Let as assume for contradiction that there exists a point $\boldsymbol{x}_0 \in \Omega$ such that $u(\boldsymbol{x}_0) \neq 0$. Without loss of generality we assume $u(\boldsymbol{x}_0) = \epsilon > 0$. As $u$ is continuous there exists an open ball around $\boldsymbol{x}_0$ of radius $\delta$, denoted $B_{\boldsymbol{x}_0}^\delta = \{\boldsymbol{x} \in \Omega \mid ||\boldsymbol{x} - \boldsymbol{x}_0||_2 < \delta\}$, such that $\forall \boldsymbol{x} \in B_{\boldsymbol{x}_0}^\delta \; u(\boldsymbol{x}) > \epsilon/2 > 0$.

Now let $\phi$ be a $\mathcal{C}^K$ function that is positive on $B_{\boldsymbol{x}_0}^\delta$ and $0$ elsewhere. Such a function can always be created by appropriately shifting and scaling $\phi(\boldsymbol{x}) = e^{-1/(1-||\boldsymbol{x}||_2^2)} \cdot \mathbf{1}_{\{||\boldsymbol{x}||_2 < 1\}}$. Its support is a closed ball $\bar{B}_{\boldsymbol{x}_0}^\delta = \{\boldsymbol{x} \in \Omega \mid ||\boldsymbol{x} - \boldsymbol{x}_0||_2 \leq \delta\}$ which is compact. Then

$$\int_\Omega u(\boldsymbol{x})\phi(\boldsymbol{x})d\boldsymbol{x} = \int_{B_{\boldsymbol{x}_0}^\delta} u(\boldsymbol{x})\phi(\boldsymbol{x})d\boldsymbol{x} > 0 \tag{23}$$

as both $u(\boldsymbol{x})$ and $\phi(\boldsymbol{x})$ are positive on $B_{\boldsymbol{x}_0}^\delta$. Thus we found a continuous function $\phi$ with compact support such that:

$$\int_\Omega u(\boldsymbol{x})\phi(\boldsymbol{x})d\boldsymbol{x} \neq 0 \tag{24}$$

Therefore $u$ is identically $0$ on $\Omega$. $\qquad \square$

To make this section self-contained, we provide the statement of the generalized divergence theorem [31].

**Theorem 2** (Divergence theorem). *Let $\Omega$ be an open set in $\mathbb{R}^M$ and let $f, g$ be continuous on $\bar{\Omega} = \Omega \cup \partial\Omega$ and continuously differentiable on $\Omega$. Then*

$$\int_\Omega \partial_i^1[f(\boldsymbol{x})]g(\boldsymbol{x})d\boldsymbol{x} = -\int_\Omega f(\boldsymbol{x})\partial_i^1[g(\boldsymbol{x})]d\boldsymbol{x} + \int_{\partial\Omega} \nu_i f(\boldsymbol{x})g(\boldsymbol{x})d\boldsymbol{x} \tag{25}$$

*where $\boldsymbol{\nu}$ is a normal unit vector to the boundary $\partial\Omega$.*

In a 1-dimensional setting, the statement of the theorem reduces to the integration by parts.

We can now prove Theorem 1.

*Proof.* Let us denote $\mathcal{E}_p = (\boldsymbol{\alpha}_p, a_p, h_p)$. Then the PDE in Equation 9 can be written as:

$$\sum_{p=1}^P a_p(\boldsymbol{x})\partial^{\boldsymbol{\alpha}_p}[h_p(\boldsymbol{x}, \boldsymbol{u}(\boldsymbol{x}))] - g(\boldsymbol{x}, \boldsymbol{u}(\boldsymbol{x})) = 0 \;\forall \boldsymbol{x} \in \Omega \tag{26}$$

The LHS is continuous as all $a_p$ and $h_p$ are smooth, $g$ is continuous, $u \in \mathcal{C}^K$, and $|\boldsymbol{\alpha}_p| \leq K \ \forall p \in [P]$. Thus we can use the fundamental lemma of calculus of variations to say that the Equation 26 is true if and only if

$$\int_\Omega \left[ \sum_{p=1}^P a_p(\boldsymbol{x}) \partial^{\boldsymbol{\alpha}_p}[h_p(\boldsymbol{x}, \boldsymbol{u}(\boldsymbol{x}))] - g(\boldsymbol{x}, \boldsymbol{u}(\boldsymbol{x})) \right] \phi(x) d\boldsymbol{x} = 0 \tag{27}$$

for all testing functions $\phi$. We transform the LHS of Equation 27 using linearity to:

$$\sum_{p=1}^P \int_\Omega a_p(\boldsymbol{x}) \partial^{\boldsymbol{\alpha}_p}[h_p(\boldsymbol{x}, \boldsymbol{u}(\boldsymbol{x}))] \phi(x) - \int_\Omega g(\boldsymbol{x}, \boldsymbol{u}(\boldsymbol{x})) \phi(\boldsymbol{x}) d\boldsymbol{x} \tag{28}$$

Let us now focus on

$$\int_\Omega a_p(\boldsymbol{x}) \partial^{\boldsymbol{\alpha}_p}[h_p(\boldsymbol{x}, \boldsymbol{u}(\boldsymbol{x}))] \phi(x) \tag{29}$$

and let us denote $\boldsymbol{\alpha}_p = (\alpha_{p1}, \ldots, \alpha_{pM})$. Then $\partial^{\boldsymbol{\alpha}_p} = \partial_1^{\alpha_{p1}} \ldots \partial_M^{\alpha_{pM}}$ and the expression can be written as

$$\int_\Omega \partial_1^{\alpha_{p1}} \ldots \partial_M^{\alpha_{pM}}[h_p(\boldsymbol{x}, \boldsymbol{u}(\boldsymbol{x}))] a_p(\boldsymbol{x}) \phi(x) d\boldsymbol{x} \tag{30}$$

Let us denote the support of $\phi$ as $\mathcal{B}$. As $\phi$ is equal to zero outside of its support, we can write the expression as

$$\int_\mathcal{B} \partial_1^{\alpha_{p1}} \ldots \partial_M^{\alpha_{pM}}[h_p(\boldsymbol{x}, \boldsymbol{u}(\boldsymbol{x}))] a_p(\boldsymbol{x}) \phi(x) d\boldsymbol{x} \tag{31}$$

Without loss of generality, let us assume that $\alpha_{p1} > 0$. By the divergence theorem, this can be rewritten as

$$- \int_\mathcal{B} \partial_1^{\alpha_{p1}-1} \ldots \partial_M^{\alpha_{pM}}[h_p(\boldsymbol{x}, \boldsymbol{u}(\boldsymbol{x}))] \partial_1^1[a_p(\boldsymbol{x}) \phi(x)] d\boldsymbol{x} \tag{32}$$

because the integral over the boundary is equal to 0

$$\int_{\partial \mathcal{B}} \nu_1 \partial_1^{\alpha_{p1}-1} \ldots \partial_M^{\alpha_{pM}}[h_p(\boldsymbol{x}, \boldsymbol{u}(\boldsymbol{x}))] a_p(\boldsymbol{x}) \phi(x) d\boldsymbol{x} = 0 \tag{33}$$

as $\phi$ has a compact support (and thus vanishes on the boundary). We can perform this operation $\alpha_{p1}$ times to shift the whole derivative $\partial_1^{\alpha_{p1}}$ to the second part of the equation and obtain

$$(-1)^{\alpha_{p1}} \int_\mathcal{B} \partial_2^{\alpha_{p2}} \ldots \partial_M^{\alpha_{pM}}[h_p(\boldsymbol{x}, \boldsymbol{u}(\boldsymbol{x}))] \partial_1^{\alpha_{p1}}[a_p(\boldsymbol{x}) \phi(x)] d\boldsymbol{x} \tag{34}$$

Then we repeat this for other derivatives and we end up with the following expression:

$$(-1)^{\alpha_{p1}} \cdot \ldots \cdot (-1)^{\alpha_{pM}} \int_\mathcal{B} h_p(\boldsymbol{x}, \boldsymbol{u}(\boldsymbol{x})) \partial_1^{\alpha_{p1}} \ldots \partial_M^{\alpha_{pM}}[a_p(\boldsymbol{x}) \phi(x)] d\boldsymbol{x} \tag{35}$$

As the integrand is zero outside of $\mathcal{B}$, this can be rewritten as:

$$(-1)^{|\boldsymbol{\alpha}_p|} \int_\Omega h_p(\boldsymbol{x}, \boldsymbol{u}(\boldsymbol{x})) \partial^{\boldsymbol{\alpha}_p}[a_p(\boldsymbol{x}) \phi(x)] d\boldsymbol{x} \tag{36}$$

or more compactly, using the functional defined in Definition 3, as:

$$\mathcal{F}(\mathcal{E}_p, \boldsymbol{u}, \phi) \tag{37}$$

Therefore Equation 28 can be written as:

$$\sum_{p=1}^P \mathcal{F}(\mathcal{E}_p, \boldsymbol{u}, \phi) - \int_\Omega [g(\boldsymbol{x}, \boldsymbol{u}(\boldsymbol{x})) \phi(\boldsymbol{x})] d\boldsymbol{x} \tag{38}$$

Thus, we proved that Equation 26 is true if and only if

$$\sum_{p=1}^P \mathcal{F}(\mathcal{E}_p, \boldsymbol{u}, \phi) - \int_\Omega [g(\boldsymbol{x}, \boldsymbol{u}(\boldsymbol{x})) \phi(\boldsymbol{x})] d\boldsymbol{x} = 0 \tag{39}$$

for all testing functions $\phi$. $\qquad \square$

Table 5: Examples of equations which are Variational-Ready

| Name | Equation | Linear | VR |
|---|---|---|---|
| Damped wave eq. with a source | $u_{tt} + \rho u_t - \kappa \nabla^2 u = g(\boldsymbol{x})$ | ✓ | ✓ |
| Gauss law | $\nabla \cdot \boldsymbol{E} = \rho/\epsilon_0$ | ✓ | ✓ |
| Burger's equation | $u_t + u u_x - \nu u_{xx} = 0$ | ✗ | ✓ |
| Navier-Stokes equations | $\boldsymbol{u}_t + (\boldsymbol{u} \cdot \nabla)\boldsymbol{u} - \nu \nabla^2 \boldsymbol{u} = -1/\rho \nabla p + \boldsymbol{g}$ | ✗ | ✓ |
| Korteweg-De Vries equation | $u_t + u_{xxx} - 6 u u_x = 0$ | ✗ | ✓ |
| Kuramoto-Sivashinsky equation | $u_t + u_{xx} + u_{xxxx} + u u_x = 0$ | ✗ | ✓ |
| Fisher's equation | $u_t - \kappa u_{xx} = r u (1 - u)$ | ✗ | ✓ |
| Liouville's equation | $u_{xx} + u_{yy} = \kappa e^{\rho u}$ | ✗ | ✓ |
| Porous medium equation | $u_t - \nabla^2(u^\kappa) = 0$ | ✗ | ✓ |
| Sine-Gordon equation | $u_{tt} - u_{xx} = -\sin(u)$ | ✗ | ✓ |

### B.3 Examples

The examples of VR-PDEs can be found in Table 5.

## C D-CIPHER

### C.1 Rewrite the inner optimization as a constrained least squares

Let us rewrite the objective in Equation 14.

$$\sum_{d=1}^{D}\sum_{s=1}^{S}\left(\sum_{p=1}^{P}\mathcal{F}(\beta_p \hat{\mathcal{E}}_p, \hat{\boldsymbol{u}}^{(d)}, \phi_s) - \int_\Omega g(\boldsymbol{x}, \hat{\boldsymbol{u}}^{(d)}(\boldsymbol{x}))\phi_s(\boldsymbol{x})d\boldsymbol{x}\right)^2 \tag{40}$$

First, let us observe that

$$\mathcal{F}(\beta_p \hat{\mathcal{E}}_p, \hat{\boldsymbol{u}}^{(d)}, \phi_s) = \int_\Omega h_p(\boldsymbol{x}, \hat{\boldsymbol{u}}^{(d)}(\boldsymbol{x}))(-1)^{|\boldsymbol{\alpha}_p|}\partial^{\boldsymbol{\alpha}_p}[\beta_p a_p(\boldsymbol{x})\phi_s(\boldsymbol{x})]d\boldsymbol{x} = \beta_p z_p^{(d,s)} \tag{41}$$

if we let $z_p^{(d,s)} \in \mathbb{R}$ be defined as

$$z_p^{(d,s)} = \int_\Omega h_p(\boldsymbol{x}, \hat{\boldsymbol{u}}^{(d)}(\boldsymbol{x}))(-1)^{|\boldsymbol{\alpha}_p|}\partial^{\boldsymbol{\alpha}_p}(a_p(\boldsymbol{x})\phi_s(\boldsymbol{x}))d\boldsymbol{x} \tag{42}$$

Moreover, if we define $w^{(d,s)} \in \mathbb{R}$ as

$$w^{(d,s)} = \int_\Omega g(\boldsymbol{x}, \hat{\boldsymbol{u}}^{(d)}(\boldsymbol{x}))\phi_s(\boldsymbol{x})d\boldsymbol{x} \tag{43}$$

we can rewrite expression 40 as

$$\sum_{d=1}^{D}\sum_{s=1}^{S}\left(\sum_{p=1}^{P}\beta_p z_p^{(d,s)} - w^{(d,s)}\right) \tag{44}$$

Now, the sum over $p$ can be written as a dot product between $\boldsymbol{z}^{(d,s)} \in \mathbb{R}^P$ and $\boldsymbol{\beta} \in \mathbb{R}^P$. We can also combine the sums over $d$ and $s$. We obtain

$$\sum_{(d,s)\in[D]\times[S]}\left(\boldsymbol{\beta}\cdot\boldsymbol{z}^{(d,s)} - w^{(d,s)}\right)^2 \tag{45}$$

which is exactly the same as the objective in Equation 15.

### C.2 Pseudocode

The pseudocode of D-CIPHER is presented in Algorithm 1.

**Algorithm 1** D-CIPHER

---

**Input:** Observed fields $\mathcal{D} = \{\boldsymbol{v}^{(d)}\}_{d=1}^{D}$, grid $\mathcal{G}$
**Input:** Symbolic regression optimization algorithm $\mathcal{O}$
**Input:** Smoothing algorithm $\mathcal{S}$
**Input:** Testing functions $\{\phi_s\}_{s=1}^{S}$
**Input:** Dictionary $\mathcal{Q} = \{\hat{\mathcal{E}}_p\}_{p=1}^{P}, \hat{\mathcal{E}}_p = (\boldsymbol{\alpha}_p, a_p, h_p)$        ▷ Step 1
**Output:** Target PDE
    $\hat{\boldsymbol{u}}^{(d)} = \mathcal{S}(\boldsymbol{v}^{(d)}) \, \forall d \in [D]$        ▷ Step 2
    initialize matrix $\boldsymbol{Z} \in \mathbb{R}^{D \times S} \times \mathbb{R}^{P}$
    $Z_p^{(d,s)} \leftarrow \int_{\Omega} h_p(\boldsymbol{x}, \hat{\boldsymbol{u}}^{(d)}(\boldsymbol{x}))(-1)^{|\boldsymbol{\alpha}_p|} \partial^{\boldsymbol{\alpha}_p}(a_p(\boldsymbol{x})\phi_s(\boldsymbol{x})) d\boldsymbol{x}$
    **procedure** LOSS($g$)
        initialize vector $\boldsymbol{w} \in \mathbb{R}^{D \times S}$
        $w^{(d,s)} \leftarrow \int_{\Omega} g(\boldsymbol{x}, \hat{\boldsymbol{u}}^{(d)}(\boldsymbol{x}))\phi_s(\boldsymbol{x}) d\boldsymbol{x}$
        $\boldsymbol{\beta} \leftarrow$ COLLIE($\boldsymbol{Z}, \boldsymbol{w}$)        ▷ Section 7
        $L = ||\boldsymbol{Z}\boldsymbol{\beta} - \boldsymbol{w}||_2^2$
        **return** $L$
    **end procedure**
    $g = \mathcal{O}(\text{LOSS})$        ▷ Step 3
    initialize vector $\boldsymbol{w} \in \mathbb{R}^{D \times S}$
    $w^{(d,s)} \leftarrow \int_{\Omega} g(\boldsymbol{x}, \hat{\boldsymbol{u}}^{(d)}(\boldsymbol{x}))\phi_s(\boldsymbol{x}) d\boldsymbol{x}$
    $\boldsymbol{\beta} \leftarrow$ COLLIE($\boldsymbol{Z}, \boldsymbol{w}$)        ▷ Section 7
    **return** $\sum_{p=1}^{P} \beta_p \hat{\mathcal{E}}_p[\boldsymbol{u}](\boldsymbol{x}) - g(\boldsymbol{x}, \boldsymbol{u}(\boldsymbol{x})) = 0$

---

### C.3 Testing functions

Testing functions should satisfy the following conditions:

1. Be sufficiently smooth (at least $\mathcal{C}^K$ for a $K^{\text{th}}$ order PDE)

2. Compact support

3. Derivatives can be computed analytically

4. Orthonormal

Conditions 1 and 2 follow directly from Definition 3. Condition 3 is necessary because we do not want to estimate the derivatives of the testing functions. Condition 4 follows from the result obtained by [42] that suggests that these functions should be a subset of an orthonormal basis of L2 space.

We use B-Splines [15] as the testing functions in our experiments because we can control their smoothness and the derivatives are easy to compute. We scale and shift them appropriately so that they are orthonormal.

Other testing functions are possible and examining them constitutes an interesting research direction. In particular, piecewise polynomials as defined by [35] or various wavelets. Ideally, we would like to choose wavelets that form an orthonormal basis for the L2 space, such as

- Shannon wavelets - smooth ($\mathcal{C}^{\infty}$) but not compact

- Meyer wavelets - smooth ($\mathcal{C}^{\infty}$) but not compact (better rate of decay than Shannon)

- Daubechies wavelets - smooth ($\mathcal{C}^K$ for a specified $K$) and compact but they do not have a closed-form expression.

Another interesting avenue of research would be to adapt the testing functions to the input data.

# D CoLLie

## D.1 Lagrangian

The problem that CoLLie is supposed to solve is a constrained least-squares optimization defined as:

$$\text{minimize } ||\boldsymbol{A}\boldsymbol{z} - \boldsymbol{b}||_2^2 \\ \text{subject to } ||\boldsymbol{z}||_1 - 1 = 0 \tag{46}$$

where $\boldsymbol{A} \in \mathbb{R}^{m \times n}$ has a full column rank, $\boldsymbol{b} \in \mathbb{R}^m$, and $\boldsymbol{z} \in \mathbb{R}^n$ for some $m, n \in \mathbb{N}$.

We consider the *Lagrangian* $L : \mathbb{R}^n \times \mathbb{R} \to \mathbb{R}$ associated with this problem [8] defined as

$$L(\boldsymbol{z}, \lambda) = ||\boldsymbol{A}\boldsymbol{z} - \boldsymbol{b}||_2^2 + \lambda(||\boldsymbol{z}||_1 - 1) \tag{47}$$

Now let us define $\hat{\boldsymbol{z}} : \mathbb{R} \to \mathbb{R}^n$ as

$$\hat{\boldsymbol{z}}(\lambda) = \underset{\boldsymbol{z} \in \mathbb{R}^n}{\arg\min} \, L(\boldsymbol{z}, \lambda) = \underset{\boldsymbol{z} \in \mathbb{R}^n}{\arg\min} \, ||\boldsymbol{A}\boldsymbol{z} - \boldsymbol{b}||_2^2 + \lambda||\boldsymbol{z}||_1 \tag{48}$$

The goal of our algorithm is to find $\lambda^* \in \mathbb{R}$ such that $||\hat{\boldsymbol{z}}(\lambda^*)||_1 = 1$. Let us define a function $q : \mathbb{R} \to \mathbb{R}$ as

$$q(\lambda) = ||\hat{\boldsymbol{z}}(\lambda)||_1 \tag{49}$$

The goal can be phrased as finding $\lambda^* \in \mathbb{R}$ such that $q(\lambda^*) = 1$.

Let us note that $\hat{\boldsymbol{z}}(0)$ is just a solution to the ordinary least squares (OLS) problem with no constraints and its norm is $q(0)$.

## D.2 Extending LARS

**Case 1.** $q(0) \geq 1$.

If we assume that $\hat{\boldsymbol{z}}$ is continuous then $q$ is also continuous. From the continuity and the fact that $\lim_{\lambda \to +\infty} q(\lambda) = 0$ and $q(0) \geq 1$ we infer that there exists a $\lambda \geq 0$ such that $q(\lambda) = 1$. Moreover, for $\lambda \geq 0$ the problem in Equation 48 is the same as in LASSO [53]. Therefore we just need to perform LASSO for different $\lambda$ and choose the one that gives the solution with L1 norm equal to 1.

To do it in practice we use Least Angle Regression (LARS) [17], a popular algorithm used to minimize the LASSO objective. It generates complete solution paths, i.e., a function $\boldsymbol{c} : \mathbb{R}_+ \to \mathbb{R}^n$ defined as

$$\boldsymbol{c}(\lambda) = \underset{\boldsymbol{z} \in \mathbb{R}^n}{\arg\min} \, ||\boldsymbol{A}\boldsymbol{z} - \boldsymbol{b}||_2^2 + \lambda||\boldsymbol{z}||_1 \tag{50}$$

which is equivalent to $\hat{\boldsymbol{z}}$ for $\lambda \geq 0$. An illustration of LARS solution paths can be seen in Figure 6. Each line corresponds to a function $c_i$ which describes the coefficient for the $i^{\text{th}}$ covariate. The paths are defined from some $\lambda_0$ where all $c_i(\lambda_0) = 0$ to $\lambda = 0$ where $\boldsymbol{c}(0) = \hat{\boldsymbol{z}}(0)$. In other words, the solution paths cover the whole range of constraints from the strictest, effectively imposing the L1 norm of $\boldsymbol{z}$ to be 0, up to no constraints, solving the OLS problem.

The solution paths from the LARS algorithm are piecewise linear and the outputs are the values of the coefficients for points $(\lambda_0 > \ldots > \lambda_n = 0)$ where the slopes change. We calculate the norm at each of these points, $||\boldsymbol{c}(\lambda_i)||_1$, and find $j \in [n]$ such that $||\boldsymbol{c}(\lambda_{j-1})||_1 < 1 \leq ||\boldsymbol{c}(\lambda_j)||_1$. As each $c_i$ is a linear function on $[\lambda_j, \lambda_{j-1}]$ and we know both $\boldsymbol{c}(\lambda_{j-1})$ and $\boldsymbol{c}(\lambda_j)$, we can effectively search for $\lambda \in [\lambda_j, \lambda_{j-1}]$ such that $||\boldsymbol{c}(\lambda)||_1 = 1$. The search can be performed by any root-finding algorithm. We use Brent's method [9].

**Case 2.** $0 < q(0) < 1$.

This is much more difficult as it corresponds to solving the problem in Equation 48 for $\lambda < 0$. The solutions given by the LARS algorithm are too small. In fact, the solution with the biggest norm is $\boldsymbol{c}(0) = \hat{\boldsymbol{z}}(0)$, the OLS solution, with norm exactly $q(0) < 1$.

To address this challenge, we propose the following heuristic. We *extend* the solution paths generated by LARS beyond $\lambda = 0$ for $\lambda < 0$. We assume that the paths will continue to be piecewise linear and that they will keep the slope they have in the last interval $[\lambda_n = 0, \lambda_{n-1}]$. Let us denote this slope as

$$\Delta c_i = \frac{c_i(0) - c_i(\lambda_{n-1})}{0 - \lambda_{n-1}} \tag{51}$$

This is graphically represented in Figure 6. Formally, these extended paths, $\bar{c} : \mathbb{R} \to \mathbb{R}$ are defined as:

$$\bar{c}_i(\lambda) = \begin{cases} c_i(\lambda), & \lambda \geq 0 \\ c_i(0) + \lambda \Delta c_i, & \lambda < 0 \end{cases} \tag{52}$$

Now, we want to find $\lambda < 0$ such that $||\bar{c}(\lambda)||_1 = 1$. To achieve this in practice, we first make the following observations.

For any $\lambda < 0$ we say that $\bar{c}_i(\lambda)$ is on the *right side* if $\bar{c}_i(\lambda)\Delta c_i \leq 0$ and we say that $\bar{c}_i(\lambda)$ is on the *wrong side* if $\bar{c}_i(\lambda)\Delta c_i > 0$. In other words, being on the wrong side just means that the path is yet to cross the x-axis if we keep decreasing $\lambda$. We can easily find $\lambda'$ such that for all $\lambda < \lambda'$ all $\bar{c}_i(\lambda)$ are on the right side (none of the paths will ever cross the x-axis).

$$\lambda' = \min \left\{ \frac{0 - c_i(0)}{\Delta c_i} \mid i \in [n] \wedge c_i(0)\Delta c_i > 0 \right\} \tag{53}$$

If $||\bar{c}(\lambda')||_1 \geq 1$ we just need to search the interval $[\lambda', 0]$ for $\lambda$ such that $||\bar{c}(\lambda')||_1 = 1$.

If $||\bar{c}(\lambda')||_1 < 1$ then we need to search $\lambda < \lambda'$. However, by definition, for all $\lambda < \lambda'$, all $c_i(\lambda)$ are on the right side. That means $||\bar{c}(\lambda)||_1$ as a function of $\lambda$ is just a linear function on the interval $(-\infty, \lambda')$. To see that, let us observe that

$$||\bar{c}(\lambda)||_1 = \sum_{i=1}^{n} |\bar{c}_i(\lambda)| = \sum_{i=1}^{n} sign(\bar{c}_i(\lambda))\bar{c}_i(\lambda) \tag{54}$$

Additionally, for $\lambda < \lambda'$ all $c_i(\lambda)$ are on the right side, so we have $sign(\bar{c}_i(\lambda)) = -sign(\Delta c_i)$. We can rewrite $||\bar{c}(\lambda)||_1$ as:

$$
\begin{aligned}
||\bar{c}(\lambda)||_1 &= \sum_{i=1}^{n} (-sign(\Delta c_i)(c_i(0) + \lambda \Delta c_i)) \\
&= -\sum_{i=1}^{n} sign(\Delta c_i)c_i(0) - \left( \sum_{i=1}^{n} sign(\Delta c_i)\Delta c_i \right) \lambda \\
&= -\sum_{i=1}^{n} sign(\Delta c_i)c_i(0) - \left( \sum_{i=1}^{n} |\Delta c_i| \right) \lambda
\end{aligned}
\tag{55}
$$

Therefore the solution can be found using the following equation

$$\lambda^* = \lambda' + \frac{1 - ||\bar{c}(\lambda')||_1}{-\sum_{i=1}^{n} |\Delta c_i|} \tag{56}$$

**Case 3.** $q(0) = 0$. In that case, we just return a precomputed solution to the problem

$$
\begin{aligned}
&\text{minimize } ||\boldsymbol{A}\boldsymbol{z}||_2^2 \\
&\text{subject to } ||\boldsymbol{z}||_1 - 1 = 0
\end{aligned}
\tag{57}
$$

which we compute by subdividing the problem into $2^n$ quadratic programs and solving each of them separately using CVXOPT algorithm [2] as described in Section 7.

### D.3 Comparison

We perform a comparison between CoLLie and an algorithm based on CVXOPT [2] as described in Section 7. CoLLie uses different procedures depending on the problem, thus for a fair comparison we generate an equal number of tests falling under Case 1 and Case 2 (two main cases) as described in D.2. To achieve that, we generate a random $m \times n$ matrix $\boldsymbol{A}$ where each entry is sampled from the standard normal distribution. We use $m = 1000$ and $n$ ranging from 2 to 7. We then generate a vector $\hat{\boldsymbol{z}} \in \mathbb{R}^n$ such that each entry is sampled from a uniform distribution on $[-0.5, 0.5]$. Then $\hat{\boldsymbol{z}}$ is normalized to have L1 norm equal to 1. We sample a number $l$ from a uniform distribution on $[-1, 3]$ and multiply $\hat{\boldsymbol{z}}$ by $l$ to obtain $\boldsymbol{z}' = l\hat{\boldsymbol{z}}$. By this procedure, we are guaranteed that cases $||\boldsymbol{z}'||_1 < 1$

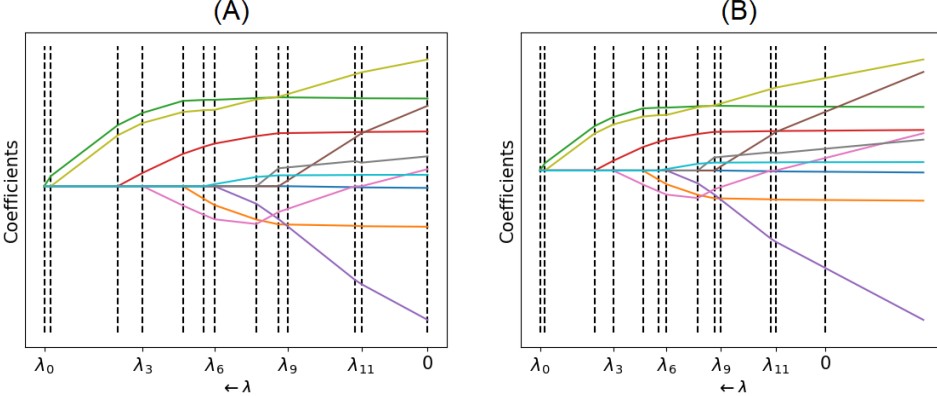

Figure 6: Panel **A** shows and example of solution paths calculated by the LARS algorithm. Panel **B** shows their extended versions as defined in Case 2 in D.2. The x-axis is reversed, so $\lambda$ decreases as it moves to the right.

and $||\boldsymbol{z}'||_1 \geq 1$ are equally likely. We then let $\boldsymbol{b} = \boldsymbol{A}\boldsymbol{z}'$. The task is then to find $\boldsymbol{z}$ with L1 norm equal to 1 that minimizes $||\boldsymbol{A}\boldsymbol{z} - \boldsymbol{b}||_2^2$. As $q(0) = ||\boldsymbol{z}'||_1 = l$, Case 1 and Case 2 are equally likely.

We perform 1000 experiments for each $n$. As losses for optimal solutions can be on widely different scales, we report the relative error between the loss obtained by CoLLie and the loss obtained by the algorithm based on CVXOPT, which seems to always find the optimal solution.

The time is measured on a single computer with an Intel Core i5-6500 CPU (4 cores) and 16GB of RAM.

# E    Experiments

## E.1    Ablated D-CIPHER

The ablated version uses the standard MSE loss with estimated derivatives and thus solves the following optimization problem:

$$\min_{g} \min_{||\boldsymbol{\beta}||_1} \sum_{d=1}^{D} \sum_{\boldsymbol{x} \in \mathcal{G}} \left( \sum_{p=1}^{P} \beta_p \hat{\mathcal{E}}_p[\boldsymbol{v}^{(d)}](\boldsymbol{x}) - g(\boldsymbol{x}, \boldsymbol{v}^{(d)}(\boldsymbol{x})) \right)^2 \tag{58}$$

where $\boldsymbol{v}^{(d)}$ is the observed field, $\mathcal{G}$ is the sampling grid, and $\hat{\mathcal{E}}_p[\boldsymbol{v}^{(d)}](\boldsymbol{x})$ requires derivative estimation.

The ablated version uses the same symbolic regression algorithm to search over closed-form $g$ and CoLLie for the inner optimization.

## E.2    Implementation

**Step 1.** For the homogeneous heat equation and Burgers' equation we use dictionary $\mathcal{Q} = \{u, \partial_t u, \partial_x u, \partial_x(u^2), \partial_x^2 u, \partial_x^2(u^2)\}$. For Kuramoto-Sivashinsky equation we use dictionary $\mathcal{Q} = \{u, \partial_t u, \partial_x u, \partial_x(u^2), \partial_x^2 u, \partial_x^2(u^2), \partial_x^3 u, \partial_x^3(u^2), \partial_x^4 u, \partial_x^4(u^2)\}$. For the damped and forced harmonic oscillator we use the dictionary $\{\partial_t, \partial_t^2\}$, and for the wave and heat equations, we use $\{\partial_t, \partial_x, \partial_t^2, \partial_t \partial_x, \partial_x^2\}$.

**Step 2.** Field estimation is performed using the Gaussian Process Regression from the Python library scikit-learn [39]. The kernel is chosen to be the RBF kernel [60] with an added White kernel to account for noise. The observed field is initially standardized by subtracting the mean and dividing by the standard deviation. Then the GaussianProcessRegressor is fitted to the data. The estimated fields are generated by predicting the values of a trained Gaussian Process on a full integration grid

and then scaling back to their original range (by multiplying by the standard deviation and adding the mean).

**Step 3.** The search over the closed-form expression is performed using the symbolic regression library gplearn [52]. We use a custom fitness function that solves the inner optimization problem in Equation 14. This inner optimization is performed by CoLLie (Section 7). The integration is performed using Riemann sums.

**Ablated version of D-CIPHER.** The derivative estimation is performed by first fitting a Gaussian process (in the same way as in Step 2) and then using the finite difference to estimate the derivative in one of the coordinates for all points in the sampling grid. To obtain higher-order derivatives, a Gaussian process is fitted again and the derivative is once again calculated using the finite difference (possibly in a different direction than the first time).

### E.3 Hyperparameters

**Gaussian process regression.** The kernel parameters of the Gaussian Process are automatically adjusted during training. The default bounds of the length scale of the RBF kernel and the noise level of the White kernel are used, i.e., $(1e-5, 1e5)$.

**GPlearn.** We do not perform parameter tuning for the gplearn library and use the same parameters as in D-CODE [42] except for the parsimony coefficient and the number of generations.

Table 6: Hyperparameters used in gplearn

| Hyperparameter | Value |
|---|---|
| population size | 15000 |
| tournament size | 20 |
| p crossover | 0.6903 |
| p subtree mutation | 0.1330 |
| p hoist mutation | 0.0361 |
| p point mutation | 0.0905 |
| generations | 20 and 30 |

The number of generations is chosen to be 30 for the damped and forced harmonic oscillator and 20 for the inhomogeneous heat and wave equations.

Please check [52] for the detailed description of these parameters.

We modify the implementation of the parsimony coefficient. The standard implementation adds to the loss the length of the equation multiplied by the parsimony coefficient. In our implementation, we increase the loss by the parsimony coefficient. This modification is performed because for different experiments we record the loss on widely different scales. To prevent tuning this parameter for every experimental setting we introduce a penalty that can work on different scales. The parsimony coefficient is chosen manually by performing experiments for a few values. The value used in the experiments is $0.05$.

The set of allowed mathematical operations is: $\{+, -, \times, \div, \sin, \exp, \log\}$

We want to emphasize that we use the same configuration of gplearn in D-CIPHER and its ablated version.

**Integration and number of testing functions.** For the damped and forced harmonic oscillator we use 10 testing functions and the integration step $0.01$. For the inhomogeneous heat and wave equations we use 100 testing functions and integrate on a grid with steps $\delta t = 0.01$ and $\delta x = 0.01$.

**Derivative estimation in the ablated version of D-CIPHER.** The Gaussian process is configured the same way as described above. The interval used in the finite difference method to estimate the derivative was chosen to be: $10^{-3}$.

## E.4 Choice of equations

Equations used in Section 8.1 are canonical equations from physics that often appear in other works about PDE discovery [47]. The homogeneous heat equation is a second-order PDE that models how heat diffuses through a region. It contains the dissipative term $\partial_x^2 u$. Burgers' equation is second-order PDE used, for instance, in fluid mechanics or nonlinear acoustics [14]. It contains the advection term $u\partial_x u$ and the diffusion term that prevents shock formation. Kuramoto-Sivashinsky equation is a fourth-order PDE used in modelling reaction-diffusion systems [27] and is known for its chaotic behavior [22].

In Section 8.2, we chose equations of physical significance that have an interesting $\partial$-free part that is not a linear combination (as discussed in Section 3). That makes them impossible to discover by the current methods. A forced and damped harmonic oscillator is a second-order ODE. Although D-CODE [42] can discover any closed-form first-order ODE, it cannot be used to discover second-order ODEs. Thus there is currently no algorithm capable of discovering this equation. Inhomogeneous heat and wave equations are second-order PDEs where the $\partial$-free part is a source (of heat and wave respectively). Moreover, the wave equation does not have the standard evolution form, as it does not involve the $\partial_t$ term. Thus even without the source term, most of the current methods cannot be applied directly to discover this equation.

## E.5 Data generation

**Homogeneous heat equation.** The fields were generated by solving the equation $\partial_t u - \theta_1 \partial_x^2 u = 0$ ($\theta_1 = 0.25$) with Neumann boundary conditions $\partial_x u(t, 0) = \partial_x u(t, X) = 0$ and an initial condition $u(0, x) = u_0(x)$, where $u_0$ is randomly sampled from a Gaussian process. The equation is solved using the implicit BTCS scheme [25] with steps $\delta t = 0.001$ and $\delta x = 0.001$. The observed field is generated by sampling $(t, x) \in [0, T] \times [0, X]$, evaluating the true field $u(t, x)$ and adding Gaussian noise. $T = 2$ and $X = 2$ are used in the experiments.

**Burger's equation.** The fields are computed by solving $\partial_t u + u\partial_x u - \theta_1 \partial_x^2 U = 0$ ($\theta_1 = 0.2$) with an initial condition $u(0, x) = u_0(x)$, where $u_0$ is randomly sampled from a Gaussian process, and with Dirichlet boundary conditions $u(t, 0) = u_0(0)$, $u(t, X) = u_0(X)$. The equation is solved the using Crank-Nicolson scheme [57] with steps $\delta t = 0.002$ and $\delta x = 0.002$. The observed field is generated by sampling $(t, x) \in [0, T] \times [0, X]$, evaluating the field $u(t, x)$ and adding Gaussian noise. $T = 2$ and $X = 2$ are used in the experiments.

**Kuramoto-Sivashinsky equation.** The solution is the same as the one used in [47]. The observed field is generated by sampling $(t, x) \in [0, T] \times [0, X]$, evaluating the field $u(t, x)$ and adding Gaussian noise. $T = 100$ and $X = 100$ are used in the experiments.

**Damped and forced harmonic oscillator.** The true fields are created by analytically solving the equation $\partial_t^2 u(t) + 2\theta_1 \theta_2 \partial_t u(t) + \theta_2^2 u(t) = \theta_3 \sin(\theta_4 t)$, where $\theta_1 = 0.5, \theta_2 = 4.0, \theta_3 = 5.0, \theta_4 = 3.0$, with random initial conditions for $u(0)$ and $\partial_t u(0)$. The observed fields are then created by sampling $t \in [0, T]$, evaluating $u(t)$, and adding Gaussian noise. $T = 2$ was used in the experiments.

**Inhomogeneous heat equation.** The true fields are computed by solving $\partial_t u(t, x) - \theta_1 \partial_x^2 u(t, x) = \theta_2 e^{\theta_3 t}$, where $\theta_1 = 0.25, \theta_2 = 1.25, \theta_3 = 1.8$, with Neumann boundary conditions $\partial_x u(t, 0) = \partial_x u(t, X) = 0$ and an initial condition $u(0, x) = u_0(x)$, where $u_0$ is randomly sampled from a Gaussian process. The equation is solved using the implicit BTCS scheme [25] with steps $\delta t = 0.001$ and $\delta x = 0.001$. The observed field is generated by sampling $(t, x) \in [0, T] \times [0, X]$, evaluating the true field $u(t, x)$ and adding Gaussian noise. $T = 2$ and $X = 2$ are used in the experiments.

**Inhomogeneous wave equation.** The true fields are computed by solving $\partial_t^2 u(t, x) - \theta_1 \partial_x^2 u(t, x) = \theta_2 e^t \sin(\theta_3 t)$, where $\theta_1 = 1.0, \theta_2 = 2.0, \theta_3 = 3.0$, with Dirichlet boundary conditions $u(t, 0) = u_0(0)$, $u(t, X) = u_0(X)$, where $u_0$ is randomly sampled from a Gaussian process and specifies the initial condition $u(0, x) = u_0(x)$. The equation is solved using the Implicit Difference Method [38] with steps $\delta t = 0.001$ and $\delta x = 0.001$. The observed field was generated by sampling $(t, x) \in [0, T] \times [0, X]$, evaluating the true field $u(t, x)$ and adding Gaussian noise. $T = 2$ and $X = 2$ were used in the experiments.

### E.6 Experimental settings

**Homogeneous heat equation**

Noise ratio ($\sigma_R$): 0.001, 0.01, 0.1

Number of samples ($D$): 10

Domain ($\Omega$): $[0, 2] \times [0.2]$

Grid ($\mathcal{G}$): $\{0, 0.07, \ldots, 2\} \times \{0, 0.07, \ldots, 2\}$

**Burgers' equation**

Noise ratio ($\sigma_R$): 0.001, 0.01, 0.1

Number of samples ($D$): 10

Domain ($\Omega$): $[0, 2] \times [0.2]$

Grid ($\mathcal{G}$): $\{0, 0.1, \ldots, 2\} \times \{0, 0.1, \ldots, 2\}$

**Kuramoto-Sivashinsky equation**

Noise ratio ($\sigma_R$): 0.001, 0.01, 0.1

Number of samples ($D$): 1

Domain ($\Omega$): $[0, 100] \times [0.100]$

Grid ($\mathcal{G}$): $\{0, 1.67, \ldots, 100\} \times \{0, 1.67, \ldots, 100\}$

**Damped and forced harmonic oscillator**

Default values are in bold.

Noise ratio ($\sigma_R$): 0.001, 0.005, **0.01**, 0.1, 0.2, 0.5

Number of samples ($D$): 1, 2, 5, **10**, 15

Domain ($\Omega$): $\mathbf{[0, 2]}$

Grid ($\mathcal{G}$): $\{0, 0.08, \ldots, 2\}$, $\mathbf{\{0, 0.1, \ldots, 2\}}$, $\{0, 0.13, \ldots, 2\}$, $\{0, 0.2, \ldots, 2\}$, $\{0, 0.4, \ldots, 2\}$

**Inhomogeneous heat equation**

Noise ratio ($\sigma_R$): 0.05, 0.1, 0.2

Number of samples ($D$): 10

Domain ($\Omega$): $[0, 2] \times [0.2]$

Grid ($\mathcal{G}$): $\{0, 0.07, \ldots, 2\} \times \{0, 0.07, \ldots, 2\}$

**Inhomogeneous wave equation**

Noise ratio ($\sigma_R$): 0.001, 0.01, 0.015

Number of samples ($D$): 10

Domain ($\Omega$): $[0, 2] \times [0.2]$

Grid ($\mathcal{G}$): $\{0, 0.07, \ldots, 2\} \times \{0, 0.07, \ldots, 2\}$

### E.7 Benchmarks

We use the implementation of PDE-FIND and WSINDy from the PySINDY library [24, 16].

An important hyperparameter in PDE-FIND and WSINDy is the library $\Theta$ used. We impose that $\mathcal{Q}$ and $\Theta \cup \{\partial_t u\}$ have the same number of elements. Moreover, solutions given by PDE-FIND and WSINDy are scaled to have the L1 norm equal to 1. Both of these measures are undertaken to ensure that RMSE error is comparable between the algorithms.

For the homogeneous heat equation and Burgers' equation we use a library

$\Theta = \{u, \partial_x, \partial_x^2 u, u \partial_x u, u \partial_x^2 u\}$.

For the Kuramoto-Sivashinsky equation, we use a library

$$\Theta = \{u, \partial_x, \partial_x^2 u, \partial_x^3 u, \partial_x^4 u, u\partial_x u, u\partial_x^2 u, u\partial_x^3 u, u\partial_x^4 u\}.$$

In the experiments, we have not optimized for the derivative-free part as it is identically equal to 0 in all equations.

### E.8 Correct functional form

To measure success probability we need to establish whether two closed-form functions match. The previous approach [42] considered their *functional forms*, i.e., expressions where all numeric constants are replaced by placeholders. By this measure, functions $\sin(3x)$ and $\sin(3.5x)$ match as they have the same functional form $\sin(Cx)$, where $C$ is a placeholder.

However, this definition is quite restrictive because functions $\sin(3x)$, $\sin(3x)+0.001$, $1.001\sin(3x)$, and $\sin(3x + 0.001)$ all have different functional forms.

We consider it an open challenge to design a good metric that would meaningfully reflect whether the correct equation is discovered. We propose the following.

For a target function $f$, we consider its *augmented form* $\tilde{f}$, defined as $\tilde{f}(x) = C_1 f(C_3 x + C_4) + C_2$, where all $C_i$ are placeholders. Then all numeric constants are turned into placeholders as well. In the end, we combine the constants. For instance, $C_1 + C_2$ becomes just $C_3$.

As an example, let us consider a function $f(x) = 1.3e^{2x}$. The augmented functional form is created in the following way:

1. Augment: $C_1 \times 1.3e^{2\times(C_3\times x+C_4)} + C_2$
2. Replace: $C_1 \times C_5 e^{C_6\times(C_3\times x+C_4)} + C_2$
3. Combine: $C_1 e^{C_3 x+C_4} + C_2$

We perform this procedure for the target function. We can now take the standard functional form of the candidate function and check whether it matches the augmented functional form of the target function, taking into account that some of the constants might not be present in the candidate expression.

To aid in this procedure, we use a Python library for symbolic mathematics, SymPy [37].

### E.9 Computation time

The average computation time for a single experiment with the damped and forced harmonic oscillator is 281 seconds with a standard error of 4.5 seconds. The average computation time for a single experiment with an inhomogeneous heat equation is 68 minutes with a standard error of 38 seconds. This time is measured on a single computer with an Intel Core i5-6500 CPU (4 cores) and 16GB of RAM.

The experiments are run simultaneously on 5 computers like the one described above. The total time for all experiments (all seeds, all equations, all experimental settings, and both versions of D-CIPHER) is 65 hours.

### E.10 Licenses

The licenses of the software used in this work are presented in Table 7

## F Additional experiments and discussion

### F.1 Sharpe-Lotka-McKendrick model

We test D-CIPHER on a population model called the Sharpe-Lotka-McKendrick model [59]. The model is described by the following equation

$$\partial_t u(t, a) + \partial_a u(t, a) + m(a)u(t, a) = 0 \tag{59}$$

Table 7: Software used and their licenses

| Software | License |
|---|---|
| gplearn | BSD 3-Clause "New" or "Revised" License |
| cvxopt | GNU General Public License |
| cvxpy | Apache License |
| sympy | New BSD License |
| scikit-learn | BSD 3-Clause "New" or "Revised" License |
| numpy | liberal BSD license |
| pandas | BSD 3-Clause "New" or "Revised" License |
| scipy | liberal BSD license |
| python | Zero-Clause BSD license |
| pysindy | MIT License |

where $m(a)$ is age-specific mortality rate. We choose $m(a) = 2e^{\theta a}$. We set $\theta = 1.5$ in the experiments.

We note that the derivative-free part of the target PDE cannot be expressed as a linear combination of functions from a finite dictionary if the parameters are not known a priori. D-CIPHER is uniquely positioned among other discovery algorithms as the only technique that can recover any mortality rate that can be represented as a closed-form expression. We show a comparison between D-CIPHER and the Ablated D-CIPHER in Table 8

Table 8: Simulation results for the Sharpe-Lotka-McKendrick model. We report the success probability of discovering the $\partial$-free part and the Average RMSE of the $\partial$-bound part. Standard deviations are shown in brackets

| Method | Sucess Probability | | Average RMSE) | |
|---|---|---|---|---|
| | $\sigma_R = 0.001$ | 0.01 | $\sigma_R = 0.001$ | 0.01 |
| D-CIPHER | 0.6 (0.15) | 0.5 (0.16) | 0.007 (0.0008) | 0.008 (0.0011) |
| Abl. D-CIPHER | 0.2 (0.13) | 0.2 (0.13) | 0.017 (0.0009) | 0.017 (0.0008) |

## F.2 Discovering systems

Discovering systems of PDEs is a much harder problem than discovering a single PDE. One of the issues is the fact that we would call *indeterminism*. It follows from the following fact. If a vector field $\boldsymbol{u}$ is a solution to two differential equations $f_1$ and $f_2$, i.e.,

$$f_1(\boldsymbol{x}, \boldsymbol{u}^{(d)}(\boldsymbol{x}), \partial^{[K]}\boldsymbol{u}^{(d)}(\boldsymbol{x})) = 0 \; \forall \boldsymbol{x} \in \Omega$$
$$f_2(\boldsymbol{x}, \boldsymbol{u}^{(d)}(\boldsymbol{x}), \partial^{[K]}\boldsymbol{u}^{(d)}(\boldsymbol{x})) = 0 \; \forall \boldsymbol{x} \in \Omega \tag{60}$$

then it is also a solution to any linear combination of these equations, i.e.,

$$\lambda_1 \times f_1(\boldsymbol{x}, \boldsymbol{u}^{(d)}(\boldsymbol{x}), \partial^{[K]}\boldsymbol{u}^{(d)}(\boldsymbol{x})) + \lambda_2 \times f_2(\boldsymbol{x}, \boldsymbol{u}^{(d)}(\boldsymbol{x}), \partial^{[K]}\boldsymbol{u}^{(d)}(\boldsymbol{x})) = 0 \; \forall \boldsymbol{x} \in \Omega \tag{61}$$

for any $\lambda_1, \lambda_2 \in \mathbb{R}$. Moreover, equations can sometimes be differentiated to yield more equations.

Let us take as an example the Cauchy-Riemann equations defined as:

$$\partial_x u_1 - \partial_y u_2 = 0 \partial_x u_2 + \partial_y u_1 = 0 \tag{62}$$

Let us assume that we have a true vector field $(u_1, u_2)$ that satisfies both equations. Then the following equations are also satisfied

$$\partial_x u_1 - \partial_y u_2 + \partial_x u_2 + \partial_y u_1 = 0$$
$$\partial_x u_1 - \partial_y u_2 - \partial_x u_2 - \partial_y u_1 = 0 \tag{63}$$

We can also differentiate the Cauchy-Riemann equations to arrive at the Laplacian equations for $u_1$ and $u_2$.

$$\partial_x^2 u_1 + \partial_y^2 u_1 = 0$$
$$\partial_x^2 u_2 + \partial_y^2 u_2 = 0 \tag{64}$$

We could also combine the first-order equations with second-order equations or consider even higher-order derivatives. Although all these equations are compatible with our vector field $(u_1, u_2)$, not all of them are equally desirable to discover. That is why we believe that in any algorithm for discovering systems of differential equations substantial expert knowledge or inductive biases have to be encoded to guide the algorithm into the right equations.

Current methods do not consider systems of equations or consider a system of equations of a very particular form. In the latter case, each equation models a derivative with respect to time of a different scalar field. The system is assumed to look like this:

$$
\begin{aligned}
\partial_t u_1 &= f_1(\boldsymbol{x}, \boldsymbol{u}(\boldsymbol{x}), \partial^{[K]}\boldsymbol{u}(\boldsymbol{x})) \\
\partial_t u_2 &= f_2(\boldsymbol{x}, \boldsymbol{u}(\boldsymbol{x}), \partial^{[K]}\boldsymbol{u}(\boldsymbol{x})) \\
&\cdots \\
\partial_t u_L &= f_L(\boldsymbol{x}, \boldsymbol{u}(\boldsymbol{x}), \partial^{[K]}\boldsymbol{u}(\boldsymbol{x}))
\end{aligned}
\tag{65}
$$

In addition, the LHS of these equations is often assumed to only contain spatial derivatives.

D-CIPHER can be used to discover some systems of differential equations if enough prior knowledge is provided in the choice of the dictionary $\mathcal{Q}$. Moreover, the discovered equations are not required to have a particular evolution form as is the case in current approaches.

Firstly, we note that D-CODE [42] has been shown to discover a system of equations that looks like Equations 65 when all equations are first-order ODEs. As D-CIPHER reduces to D-CODE when applied to first-order ODEs, it is also capable of discovering such a system.

We demonstrate that D-CIPHER is able to discover both Cauchy-Riemann equations if we use two different dictionaries. Each of the dictionaries yields a different equation. Additionally, it is a well-known fact that if a vector field satisfies Cauchy-Riemann equations then the constituent scalar fields are harmonic, i.e., they satisfy Laplace's equation. Based on the same dataset we are also able to discover both Laplace's equations given another set of two different dictionaries. We note that Laplace's equation does not contain $\partial_x$ term, so most of the current methods cannot be directly applied to discover this equation. The results are presented in Figure 7. The dictionaries used to discover each of the equations are the following.

For $\partial_x u_1 - \partial_y u_2 = 0$ we use $\mathcal{Q}_1 = \{\partial_x u_1, \partial_y u_2, \partial_x^2 u_1, \partial_y u_2\}$

For $\partial_x u_2 + \partial_y u_1 = 0$ we use $\mathcal{Q}_2 = \{\partial_x u_2, \partial_y u_1, \partial_x^2 u_2, \partial_y^2 u_1\}$

For $\partial_x^2 u_1 + \partial_y^2 u_1 = 0$ we use $\mathcal{Q}_3 = \{\partial_x u_1, \partial_y u_1, \partial_x^2 u_1, \partial_y^2 u_1\}$

For $\partial_x^2 u_2 + \partial_y^2 u_2 = 0$ we use $\mathcal{Q}_4 = \{\partial_x u_2, \partial_y u_2, \partial_x^2 u_2, \partial_y^2 u_2\}$

In the experiments, we have not optimized for the derivative-free part as it is identically equal to 0 in all equations.

### F.3   Increasing the size of the dictionary $\mathcal{Q}$

We test D-CIPHER on the Sharpe-Lotka-McKendrick model (Equation 59) with different dictionaries. We start with a small dictionary $\mathcal{Q}_1 = \{\partial_t u, \partial_a u\}$, and we create every new dictionary from the previous one by adding one more extended derivative. The final dictionary contains 10 elements, $\mathcal{Q}_{10} = \{\partial_t u, \partial_a u, \partial_a^2 u, \partial_t^2 u, \partial_t \partial_a u, \partial_a(u^2), \partial_a^2(u^2), \partial_t(u^2), \partial_t^2(u^2), \partial_t(u^3), \partial_a(u^3)\}$. The Average RMSE of the $\partial$-bound part is shown in Figure 8. We do not observe any increase in average error. Note that in these experiments, we just focus on the $\partial$-bound part and do not optimize the $\partial$-free part. The relationship between the computation time and the size of the dictionary is represented in Figure 9.

### F.4   Computational complexity

We want to emphasize that PDE discovery is not a time-critical application (usually this process is performed manually by scientists) and we believe D-CIPHER's computation time is acceptable for such a task. In this section, we describe which parts of the algorithms are most computationally intensive.

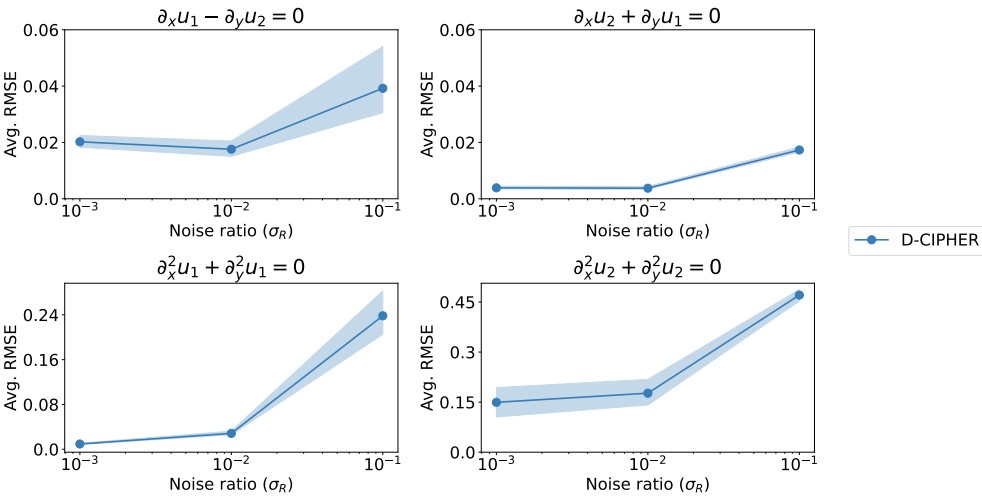

Figure 7: Simulation results for both Cauchy-Riemann equations and two Laplace's equations. We report the Average RMSE of the $\partial$-bound part in different noise settings

.

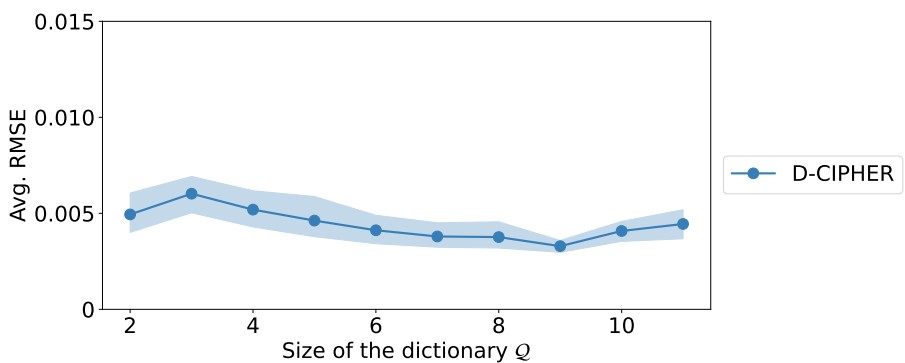

Figure 8: Simulation results for Sharpe-Lotka-McKendrick model. We report the Average RMSE of the $\partial$-bound part for different sizes of the dictionary $\mathcal{Q}$

.

Computation in D-CIPHER is performed in Step 2 and Step 3.

**Step 2.** Computational complexity of Step 2 depends on the choice of the smoothing algorithm. We want to emphasize that the user can use any smoothing algorithm based on their domain knowledge and experience, including spline regression, LOWESS, and Kalman filters. Gaussian Process has time complexity $\mathcal{O}(n^3)$ where $n$ is the number of data points in a grid $\mathcal{G}$. D-CIPHER is specifically designed to work for sparse and noisy data, so we have not encountered major computational issues while performing Gaussian process regression. We also note that significant progress has been made in adapting Gaussian processes for datasets with many data points [28].

**Step 3.** D-CIPHER consists of two optimization loops. The outer optimization is performed by a symbolic regression algorithm (in our case genetic programming). The inner optimization is performed by CoLLie (Section 7). Searching through a space of closed-form expression requires testing many candidate equations. D-CIPHER is designed to work with many different algorithms for symbolic regression and it is advised to choose an algorithm that can search through this space most efficiently.

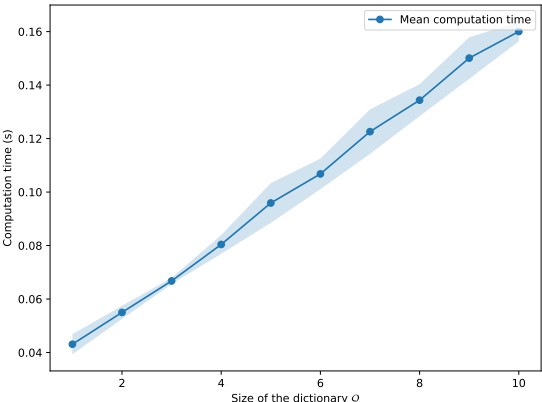

Figure 9: Computation time of the experiments in Appendix F.3 The plot shows how the size of the dictionary $\mathcal{Q}$ influences the computation time

CoLLie was specifically designed to solve the optimization problem as quickly as possible with a minor accuracy trade-off (see Figure 2). It is based on LARS which has time complexity $\mathcal{O}(mn^2)$ [17] where $m$ is the number of samples and $n$ is the number of features. In our case, $n = P$ is usually small as it corresponds to the size of the dictionary and $m = SD$. In our experiments, $S$ was set up to 100 and $D$ up to 10. Overall LARS is performed very quickly. The additional steps in CoLLie require only a few arithmetical operations (linear in $n$) and a possible root searching of a single variable function that is efficiently implemented using Brentq algorithm [9].

Other important parts of the algorithm are the numerical integrations. One such integration is performed at the beginning of the algorithm to compute the matrix $\boldsymbol{Z}$ (see Algorithm 1). It does not contribute much to the computation time as it is performed only once. The other integration is performed for each candidate equation to compute vector $\boldsymbol{w}$. Also, substantial time is spent on computing the values of the candidate function $g$ used in the integration. Fortunately, both of these operations can be implemented as vectorized operations which are designed to run very efficiently on modern hardware. We also discourage overly long equations for $g$ (check the discussion in Appendix E.3) to limit the number of operations performed.

### F.5   Challenges of derivative estimation

One of the advantages of D-CIPHER compared to other methods is the use of the variational formulation of PDEs that allows it to circumvent derivative estimation. This is important as derivative estimation is challenging, especially in noisy settings with infrequent sampling. The problem becomes more pronounced the higher the order of the derivative. To demonstrate these issues, we perform a series of synthetic experiments.

**Qualitative study.** First we qualitatively show how challenging the task of derivative estimation is. We generate an observed trajectory for the damped and forced harmonic oscillator. Then we estimate this trajectory using both Guassian Process regression and Spline regression. As shown in Figure 10 (Panel **A**), the estimated trajectories are very close to the true trajectory. Then we estimate the first derivative (Panel **B**) and the second derivative (Panel **C**). We show the standard finite difference methods as well as derivative estimation techniques using Spline regression and Gaussian Process regression. In both cases we see that the estimated derivatives do not match the ground truth (calculated analytically) as closely as in Panel **A**. Moreover the mismatch for the second derivative seems to be bigger than for the first derivative.

**Quantitative study.** We investigate this relation quantitatively for the damped and forced harmonic oscillator and the wave equation. For the oscillator we generate an observed trajectory and then we estimate its derivatives, up to the fourth order using both finite difference and Gaussian Processes. We then compare the derivatives with the analytically calculated ground truths and measure root

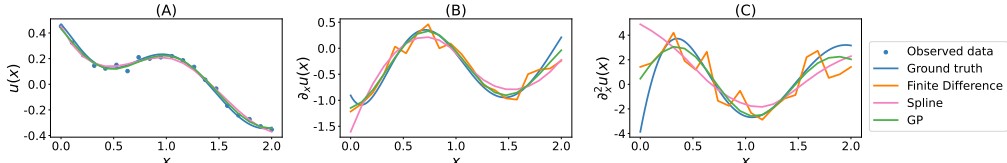

Figure 10: Panel **A** shows that estimation of the true trajectory can be performed successfully by both Spline regression (Spline) and Gaussian Process regression (GP). Panel **B** shows the estimated first derivative and Panel **C** shows the estimated second derivative. We observe that the higher the order the less accurate is the estimate.

mean squared error. The results are shown in Figure 11 (Panel **A**). We can see that the error increases the higher the order of the derivative. For the wave equation, we perform a similar experiment but this time we estimate different mixed derivatives. We consider any mixed derivative $\partial_t^i \partial_x^j$, where $i, j \in \{0, 1, 2\}$. We demonstrate the results in Figure 11 (Panel **B**). We observe that the error increases the higher the order of the derivative $(i + j)$.

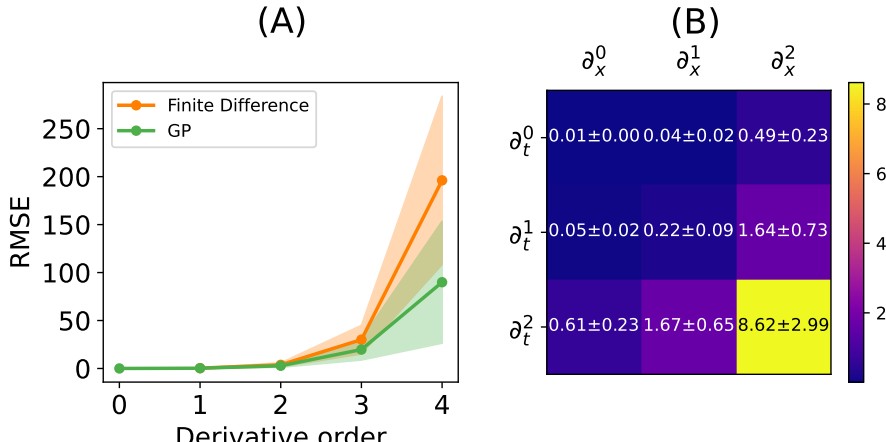

Figure 11: Panel **A** demonstrates the error between the estimated derivative and the ground truth increases with the order of the derivative (performed for the damped and forced harmonic oscillator). Panel **B** shows that the same happens for the wave equation. The $(i.j)$ entry of the heatmap should be interpreted as the RMSE between the estimated derivative $\partial_t^i \partial_x^j$ and the ground truth.

## F.6 Challenges of PDE discovery and how we address them

PDE discovery is a very difficult task with many challenges. In Table 9 we summarize some of them and describe how our work addresses them.

## F.7 Significance of the new notions

In this section, we want to justify that the new notions we introduce (evolution assumption, linear combination form, derivative-bound part, derivative-fee part, Variational-Ready PDEs) are important theoretical contributions that help us understand the landscape of different PDEs from the machine learning perspective.

The definitions we introduce let us characterize different classes of PDEs. These new notions complement the standard recognized PDE classes such as semi-linear, quasilinear, hyperbolic, etc. These standard classes were introduced predominantly to characterize the solving techniques or the properties of the solutions, whereas the notions we introduce relate to the difficulty of discovering such equations from data.

Table 9: Some challenges of PDE discovery and how we address them

| Challenge | Novel contribution (how we address the challenge) |
|---|---|
| Noisy measurements | We use the variational loss function (Equation 11) |
| Large diversity of PDEs | We allow any closed-form $\partial$-free part and require no evolution assumption (Section 3) |
| Learning compact equations | We penalize long $\partial$-free parts and use L1 normalization for the $\partial$-bound part (Section 6, Appendix E.3 |
| Efficient search | We develop a quick algorithm, CoLLie, for the inner optimization (Section 7 |
| Chaotic systems | We do *not* estimate the initial conditions [11] or perform forward time stepping [29] which are computationally unstable for chaotic systems |

**Significance of linear combination form and the evolution assumption.** In Table 10 we demonstrate how the presence of the two assumptions, the linear combination form (LC) and the evolution assumption (EA), influences the optimization problem. We see that with both assumptions, the problem is relatively straightforward and reduces to sparse linear regression. With only one of the assumptions present the problem becomes more difficult. With neither of these assumptions, the problem becomes very difficult and requires some other assumptions. D-CIPHER does not make either of these assumptions but it assumes the PDE to be of the form described by Equation 13.

**Significance of $\partial$-bound part, $\partial$-free part and Variational-Ready PDEs.** The difficulty of derivative estimation has been one of the main challenges of PDE discovery. The variational formulation allows to circumvent derivative estimation and thus is more robust to noisy data. Previously, the variational formulation has been applied only to a subset of equations in a linear combination form and with the evolution assumption. We observed that any restrictions that the variational formulation might put on the equation come from the terms containing the derivatives. Thus we define the derivative-bound part and the derivative-free part of the PDE due to their significance for the variational formulation. That allows us to define Variational-Ready PDEs as currently the broadest class of PDEs that admit the variational formulation. We believe it is an important contribution as methods requiring derivative estimation underperform in settings with high noise. This definition outlines the current limits of any method that circumvents derivative estimation in that way.

Table 10: This table demonstrates how the presence of the two assumptions, the linear combination form (LC) and the evolution assumption (EA), influences the optimization problem.

| LC | EA | Equation form | Optimization problem | Examples |
|---|---|---|---|---|
| Yes | Yes | $\partial_t u_j = \sum_{p=1}^{P} \theta_p f_p(x, \boldsymbol{u}(\boldsymbol{x}), \partial^{[K]} \boldsymbol{u}(\boldsymbol{x}))$ | Relatively easy. Can be formulated as finding sparse solution to linear least squares (similar to ridge regression) | [10, 47] |
| Yes | No | $\sum_{p=1}^{P} \theta_p f_p(x, \boldsymbol{u}(\boldsymbol{x}), \partial^{[K]} \boldsymbol{u}(\boldsymbol{x})) = 0$ | Medium difficulty. Can be formulated as $P$ separate linear least squares problems as above | [23] |
| No | Yes | $\partial_t u_j = g(\boldsymbol{x}, \boldsymbol{u}(x))$ | Medium difficulty. Find $g$ using symbolic regression | [42] |
| No | No | $f(\boldsymbol{x}, \boldsymbol{u}(\boldsymbol{x}), \partial^{[K]} \boldsymbol{u}(\boldsymbol{x})) = 0$ | Very difficult. Requires other assumptions (in this work, Equation 13) | D-CIPHER |

## F.8   Comparison with D-CODE

To clarify our contributions, we compare D-CIPHER to D-CODE [42] which we think is closest in spirit to the algorithm we developed as both of them allow any closed-form derivative-free part and

use variational formulation to circumvent derivative estimation. We also note how the new notions we introduce help us compare the two works.

Algorithm presented in D-CODE [42] can discover any first-order explicit closed-form ODE, i.e., an equation of the form.

$$\partial_t u_j(t) = g(\boldsymbol{u}(t), t)$$

where $g$ is a closed-form function. It uses the variational formulation of ODEs to circumvent derivative estimation.

Table 11: Comparison between D-CIPHER and D-CODE [42]

| Property | D-CODE [42] | D-CIPHER |
|---|---|---|
| Applicable to PDEs | No | Yes |
| Higher-order derivatives | No (only first-order) | Yes |
| Requires evolution assumption | Yes | No |
| Derivative-bound part | Fixed: $\partial_t u_j$ | Learned: $\sum_{p=1}^{P} \beta_p \hat{\mathcal{E}}_p[\boldsymbol{u}]$ |
| Derivative-free part | Any closed-form function | Any closed-form function |
| Derivative estimation | No | No |

D-CODE [42] can be considered a special case of D-CIPHER. We can recover it from D-CIPHER by choosing the dictionary to contain only one element, i.e., $\mathcal{Q} = \partial_t u_j$.

As the derivative-bound part is fixed, every ODE of that form admits the variational formulation. This is not true for PDEs as there are derivative-bound parts that might prohibit the variational formulation. Thus D-CIPHER required careful consideration of the appropriate class of equations to search over.

As D-CIPHER needs to find both the $\partial$-free part (function $g$) and the $\partial$-bound part, the optimization problem is much more complicated. That is why we restrict the derivative-bound part of the PDE to be spanned by terms from the pre-specified dictionary and develop an efficient optimization algorithm, CoLLie. We emphasize that, as is the case for [42], we do not put any constraints on the derivative-free part of the PDE, apart from it being closed-form.

### F.9 Error bounds

While we would like to have error bounds for the discovered PDE, we note that the problem we solve is significantly more difficult than the one considered in other works. The space of PDEs we consider is much more complex than the space of PDEs in a linear combination form. In other works (e.g., [47], [35]), the PDE is basically a vector in $\mathbb{R}^P$ and the discovery task is mostly reduced to finding a sparse enough vector that approximately solves a certain linear equation. Of course, there is a lot of literature that aids in establishing error bounds in such problem settings. However, D-CIPHER searches over a space $\mathbb{R}^P \times CFE(M + N)$, where $CFE(M + N)$ is a space of closed-form expressions in $M + N$ variables. This space is combinatorial in the functional form and continuous in real constants. This makes it very challenging to derive any error bounds.

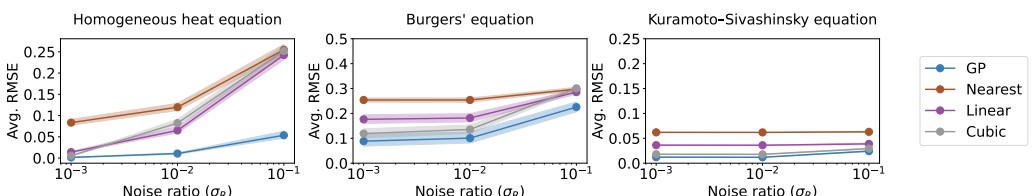

Figure 12: Comparison of different estimation algorithms that can be used in D-CIPHER. GP - Gaussian Process regression, Nearest - Nearest point interpolation, Linear - Linear interpolation, Cubic - Cubic interpolation.

### F.10 Comparison between different estimation algorithms

Step 2 of D-CIPHER requires estimating the fields. We emphasize that any choice of reconstruction algorithm can be used, and it should be chosen based on the application and domain knowledge. In our experiments, D-CIPHER is implemented using Gaussian Process regression [60]. In this section, we investigate other common interpolation algorithms. We implement D-CIPHER with different estimation algorithms. In particular, we compare Gaussian Process (GP) against: Nearest point interpolation (Nearest), Linear interpolation, and Cubic interpolation (Cubic). The implementation details of these three algorithms can be found in the scipy [56] documentation. The results are presented in Figure 12. We see that the estimation algorithms that produce smoother functions (GP, Cubic) tend to give better results.

### F.11 How comprehensive does the dictionary $\mathcal{Q}$ need to be?

Table 12 shows 17 different differential equations and what terms they need in a dictionary. All of them can be discovered with a dictionary of size of 10, $\mathcal{Q} = \{\partial_t u, \partial_x u, \partial_t\partial_x u, \partial_t^2 u, \partial_x^2 u, \partial_t\partial_x^2, \partial_x^3 u, \partial_x^4 u, \partial_x(u^2), \partial_x^2(u^2)\}$, and 9 of those equations can be described with a dictionary containing just 4 terms, $\mathcal{Q} = \{\partial_t u, \partial_x u, \partial_t^2 u, \partial_x^2 u\}$. It is thus likely that such small dictionaries are sufficient to discover most of the well-known equations. Many differential equations have similar derivative-bound parts and differ by derivative-free parts. Being able to discover any closed-form derivative-free part is what makes D-CIPHER stand out.

Table 12: This table lists various Variational-Ready PDEs (some of which are discussed in the paper) and shows which extended derivative terms they require. All of them can be discovered with a dictionary of size of 10, and 9 of them can be described with a dictionary containing just 4 terms. It is thus likely that such small dictionaries are sufficient to discover most of the well-known equations.

| Equation | $\partial_t u$ | $\partial_x u$ | $\partial_t\partial_x u$ | $\partial_t^2 u$ | $\partial_x^2 u$ | $\partial_t\partial_x^2 u$ | $\partial_x^3 u$ | $\partial_x^4 u$ | $\partial_x(u^2)$ | $\partial_x^2(u^2)$ |
|---|---|---|---|---|---|---|---|---|---|---|
| Heat equation | ✓ | | | | ✓ | | | | | |
| Wave equation | | | | ✓ | ✓ | | | | | |
| Burger's equation | ✓ | | | | ✓ | | | | ✓ | |
| SLM model | ✓ | ✓ | | | | | | | | |
| Damped harmonic oscillator | ✓ | | | ✓ | | | | | | |
| Kuramoto-Sivashinsky equation | ✓ | | | | ✓ | | | ✓ | ✓ | |
| Benjamin-Bona-Mahony equation | ✓ | ✓ | | | | ✓ | | | ✓ | |
| Boussinesq equation | | | | ✓ | ✓ | | | ✓ | | ✓ |
| Chafee-Infante equation | ✓ | ✓ | | | | | | | | |
| Damped wave equation | ✓ | | | ✓ | ✓ | | | | | |
| Fisher's equation | ✓ | ✓ | | | | | | | | |
| Hunter-Saxton equation | | | ✓ | | ✓ | | | | | ✓ |
| Klein-Gordon equation | | | | | ✓ | | | | | |
| Korteweg-De Vries equation | ✓ | | | | | | ✓ | | ✓ | |
| Liouville's equation | | | | ✓ | ✓ | | | | | |
| Sine-Gordon equation | | | | ✓ | ✓ | | | | | |
| Sinh-Gordon equation | | | ✓ | | | | | | | |

### F.12 How to take advantage of the observed derivatives?

D-CIPHER can make use of the observed derivatives (if they are available) by adapting the dictionary. Consider a setting with a dictionary $\mathcal{Q} = \{\partial_t u, \partial_x u, \partial_t\partial_x u, \partial_t^2 u, \partial_x^2 u\}$. If we happen to have the measurements of $\partial_t u$ then we can introduce a new variable $v = \partial_t u$ and change the dictionary to $\mathcal{Q} = \{v, \partial_x u, \partial_x v, \partial_t v, \partial_x^2 u\}$. Note, we have performed experiments where the dictionary contains more than one dependent variable in Appendix F.2. With observed derivatives, we can also enlarge the space of Variational-Ready PDEs by allowing $g$ ($\partial$-free part) to depend on $v$ as well.

### F.13 Impact on real-world problems

D-CIPHER is especially useful in discovering governing equations for systems with more than one independent variable. For instance, spatiotemporal data or temporal data structured by age or size.

In particular, we envision D-CIPHER to be useful in modeling spatiotemporal physical systems, population models, and epidemiological models.

**Spatiotemporal physical systems.** D-CIPHER may prove useful in discovering equations governing the oceans or the atmosphere. For instance, some places actively add or remove $CO_2$ from the atmosphere. These "sources" and "sinks" are likely to be described by a $\partial$-free part which D-CIPHER is specially equipped to discover. Similarly, with the ocean temperature where $\partial$-free part can describe a heat source. Another area of application can be modeling seismic waves across the earth's crust. Here the $\partial$-free part can describe the vibration source (e.g., an earthquake).

**Population models.** Population models can be used in agriculture to determine the harvest or for pest control to predict their impact on the crop. They have also been used in environmental conservation to model the population of endangered species. Population models have also been used in modeling the growth of cells to better understand tumor growth. Moreover, understanding the evolution of a population pyramid for a specific country may prove invaluable in ensuring its economic stability. As in all these scenarios, the rates of growth and mortality are likely to be described by $\partial$-free part, D-CIPHER is uniquely positioned to discover such equations as an aid to human experts.

**Epidemiological models.** Epidemiological models are crucial during a pandemic for better planning and interventions. For many diseases, the rates of mortality and infection are age-dependent. Thus, modeling the spread of disease using PDEs (rather than ODEs) might provide superior results.

### F.14 D-CIPHER in practice

Below we discuss the things to consider while using D-CIPHER.

**The order of the differential equation.** One of the first considerations should be to choose the order of the differential equation $K$. For many dynamical systems, $K = 2$ is sufficient unless we expect very complicated behavior. Then, considering $K = 3$ or even $K = 4$ may be warranted. Note, that we show that D-CIPHER can discover a fourth-order PDE (Kuramoto-Sivashinsky equation) in Section 8.

**Homogeneous equations.** Before searching through the whole space of closed-form $g$ (derivative-free parts), we can consider whether the equation we want to discover may be homogeneous. These experiments on the restricted search space can provide quick insights before searching through all closed-form derivative-free parts.

**Terms in a dictionary.** For a given order of a differential equation $K$, it is a good idea to include all standard differential operators up to order $K$ acting on all the variables. For instance, for $K = 2$ and $M = 1 + 1$ we could choose $\mathcal{Q} = \{\partial_t u, \partial_x u, \partial_t \partial_x u, \partial_t^2 u, \partial_x^2 u\}$. That allows to cover all linear PDEs with constant coefficients up to that order. To allow for non-linear PDEs we can include a term like $\partial_t(u^2) = 2u\partial_t u$ that often describes advection (as in Burger's equation).

**Dictionary steering when dealing with many dependent variables.** When we deal with a system of PDEs rather than a single PDE choosing a dictionary is increasingly important. As we explain in Appendix F.2, discovering whole systems of PDEs is very challenging and D-CIPHER is not designed to do so out of the box. However, we show how that can be done in certain situations. We can steer what kind of equations are discovered by choosing the terms in the dictionary.

**Estimation algorithm.** Estimation algorithms make different assumptions on the data-generating process and should be chosen based on domain expertise. As we show in Appendix F.10, algorithms that produce smoother functions, such as Gaussian Process regression and cubic spline interpolation, tend to have good results. We can consider the advantages and disadvantages of these methods. For instance, Gaussian Process regression works very well for smooth signals. However, it is computationally intensive and might not perform well if the signal is not smooth enough (it has abrupt changes). Spline interpolation, on the other hand, is faster and more appropriate for less smooth signals, but it might introduce certain unwanted artifacts because of using cubic polynomials to interpolate the data.

