# OpenReview forum: "D-CIPHER: Discovery of Closed-form Partial Differential Equations"
_NeurIPS.cc/2023/Conference — NeurIPS 2023 poster_

### Official Review · Reviewer_ubhm · 2023-07-06

**Soundness:** 3 good
**Presentation:** 3 good
**Contribution:** 3 good
**Rating:** 6
**Confidence:** 3

**Summary:**

The paper falls in the realm of data driven discovery of dynamical systems, PDEs to be specific. It proposes a framework for a class of PDEs which are termed as variation ready PDEs. These are claimed to be less restrictive than existing methods, which make stronger assumptions on the form of the PDE to be discovered and hence can't recover a significant population. A new optimization scheme, novel loss function and empirical evidence is provided to support the claims made in the paper.

**Strengths:**

In general the paper is well written, the scope of the paper is well thought.

Notations are clear and introduced properly.

It address an important problem, the clear listing of challenges in the introduction is particularly impressive.

The variational loss function is novel

**Weaknesses:**

My major concern is regarding the way the problem is setup before the solution is proposed. Some of the terms introduced here although intuitive lack enough insight to make them more convincing.

Section 7 is too short in the main paper to determine any novelty in the optimization scheme, this definitely needs to be presented better in the main paper, as this is claimed as a contribution.

One of the aspects mentioned in the paper is that of robustness from noisy or infrequent observations, I don't have see any evidence to support such claims.



**Questions:**

1) Can authors discuss the effect of the choice of test functions as B-splines? What about other options, how do they impact the results. A discussion on this ground would be helpful.
2) Can authors shed light on the robustness of the proposed framework with respect to noisy and/or infrequent observations. These if theoretical can be taking into account sampling frequency, SNR, etc.
3) How about parameterizing the test functions and making them learnable? Has this been tried, if not what do the authors feel in this regard. While test functions are fairly restrictive, there might still be a way to learn then or at the very least choosing them from a dictionary.

**Limitations:**

I don't see direct potential negative social impact. However, since this is a paper in the direction of ML for science and adjacent domains, it is quite possible that these methods when fully developed will have major impact. It will be good to have a word of caution regarding this, to make sure we acknowledge the vast amount of domain expertise already available in all scientific fields and not use ML as a tool to replace all human knowledge.

Authors have acknowledged technical limitations appropriately.

---

> ### Author Rebuttal · Authors · 2023-08-09
>
> Dear Reviewer ubhm,
>
> We appreciate your thorough feedback and encouraging comments. We summarize the improvements we have made to the paper based on your review and we answer your questions below.
>
> ### Actions taken
> 1. Elaborated on the requirements and other choices of testing functions in Appendix C.4.
> 2. Provided more details of CoLLie in Section 7 and revised the contributions section.
> 3. Added a discussion on the broader impact in Appendix F.
>
> ### Problem setup
> We are glad that you found the terms we define in Section 3 intuitive. We elaborate on their significance in Appendix F.7.
>
> Table 10, in particular, provides a concise summary of how the *evolution assumption* and the *linear combination* assumption impacts the optimization problem. As we explain in Section 3, we introduce *derivative-bound* and *derivative-free* parts of a PDE because they allow relaxing the two previously mentioned assumptions while still allowing for variational formulation (as not every PDE admits a variational formulation). This comes from a crucial observation that *no additional constraints* need to be put on the terms without derivatives to admit variational formulation.
>
> As we write in Appendix F.7,  we believe the new notions complement the standard recognized PDE classes such as semi-linear, quasilinear, hyperbolic, etc. These standard classes were introduced predominantly to characterize the solving techniques or the properties of the solutions, whereas the notions we introduce relate to the difficulty of discovering such equations from data. Therefore we believe they offer substantial insight into this area.
>
> ### CoLLie
> We have added the following description of CoLLie to Section 7 and we modified the wording of the contribution section to indicate that CoLLie is a relatively minor contribution, compared with the main contribution of the paper, i.e., introducing new notions to describe PDEs from a discovery perspective, proposing a new general class of PDEs, a novel objective function, and a discovery algorithm.
>
> **Added to Section 7:**
>
> We observe that this optimization problem is related to the one encountered in LASSO. Denote $\mathbf{z}\_0$ the solution that minimizes $||\mathbf{A}\mathbf{z}-\mathbf{b}||\_2^2$  (no constraints). If $||\mathbf{z}\_0|| > 1$ the problem is equivalent to finding $\lambda$ (in the Lagrangian form of LASSO, Equation 48) such that the LASSO solution has the norm 1. Least Angle Regression (LARS) is a popular algorithm used to minimize the LASSO objective that computes complete solution paths. These paths show how the coefficients of the solution change as $\lambda$ moves from $0$ to $\lambda_{max}$ (from no constraints to effectively imposing the solution to be $\mathbf{0}$). See Figure 6 in Appendix D2. CoLLie uses these solution paths to calculate the exact solution to the optimization problem. The case $0 < ||\mathbf{z}_0|| < 1$ is harder as it corresponds to $\lambda < 0$. CoLLie addresses this challenge by *extending* the solution paths generated by LARS beyond $\lambda=0$ for $\lambda<0$. We assume that the paths continue to be piecewise linear and keep their slope (Fig. 6, App. D2). CoLLie then uses these assumptions to efficiently find an approximate solution. We provide a detailed description of CoLLie in Appendix D.
>
> ### Robustness to noisy and infrequent observations
> As we show in Appendix F.5, estimating the derivatives is challenging and it becomes more difficult the higher the order of the derivative (see Figure 10). We thus show empirically that our method, which circumvents derivative estimation, performs better than the alternatives in scenarios with varying noise levels (Figure 2, Figure 3, Figure 4, Table 3, Table 8). We also show the impact of the frequency of observations and the number of samples in Figure 3. In all scenarios, our method is more robust than the alternative. We discuss the challenge of establishing theoretical error bounds in Appendix F.9.
>
> ### Testing functions
> Testing functions should satisfy the following conditions:
> 1. Be sufficiently smooth (at least $\mathcal{C}^K$ for a $K^{th}$ order PDE)
> 2. Compact support
> 3. Derivatives can be computed analytically
> 4. Orthonormal
>
> Conditions 1 and 2 follow directly from Definition 3. Condition 3 is necessary because we do not want to estimate the derivatives of the testing functions. Condition 4 follows from the result obtained in [1] that suggests that these functions should be a subset of an orthonormal basis of L2 space.
>
> Other testing functions are possible and examining them constitutes an interesting research direction. In particular, piecewise polynomials as defined in [2] or various wavelets. Ideally, we would like to choose wavelets that form an orthonormal basis for the L2 space such as
> - Shannon wavelets - smooth ($\mathcal{C}^{\infty}$) but not compact
> - Meyer wavelets - smooth ($\mathcal{C}^{\infty}$) but not compact (better rate of decay than Shannon)
> - Daubechies wavelets - smooth ($\mathcal{C}^{K}$ for a specified $K$) and compact but they do not have a closed-form expression.
>
> Another interesting avenue of research would be to adapt the testing functions to the input data.
>
> ### Impact
> We have added a section that acknowledges that D-CIPHER is not designed to or capable of replacing human experts in scientific discovery and should be employed as a support tool as a part of a much broader scientific process.
>
> **References**
> 1. Qian, Z., Kacprzyk, K. & van der Schaar, M. D-CODE: Discovering Closed-form ODEs from Observed Trajectories. ICLR (2022).
> 2. Messenger, D. A. & Bortz, D. M. Weak SINDy for partial differential equations. Journal of Computational Physics (2021).
>
> Thank you once again for dedicating your time to reviewing our paper. We hope our responses have been satisfactory in addressing your queries. If any aspects still require additional explanation or if you have further questions, please let us know. We are eager to provide the necessary explanations.

---

> > ### Comment · Reviewer_ubhm · 2023-08-18
> >
> > Thank you for the detailed response, and proposed update. I will keep my score.

---

> > > ### Author Response · Authors · 2023-08-19
> > >
> > > Dear Reviewer ubhm,
> > >
> > > We want to express our sincere gratitude for the time and effort you dedicated to evaluating our paper and the rebuttal. We are pleased by your continued positive assessment of our work. Your insightful comments have undeniably contributed to the enhancement of our paper.
> > >
> > > Kind regards,
> > >
> > > Authors of Submission13865

---

### Official Review · Reviewer_Znhz · 2023-07-16

**Soundness:** 3 good
**Presentation:** 4 excellent
**Contribution:** 2 fair
**Rating:** 6
**Confidence:** 3

**Summary:**

This paper proposes a framework (D-CIPHER) to discover closed-form PDEs and ODEs. The framework is more general than some of the previously existing methods, and in particular can handle a class of PDEs defined as variational-ready PDEs in the paper. The empirical experiments evaluated the discovery performance on synthetic data for a set of different equations (showing both comparisons with existing methods on discovering linear combinations and results on discovering more challenging equations).

**Strengths:**

Originality: From what I could tell, this paper proposes an original framework to discover broader classes of PDEs and ODEs. That being said, I'm not familiar with the literature in the area so I cannot fully speak to the originality.

Quality: The empirical experiments, even though synthetic, appear to be thoughtfully designed and demonstrate notable improvements.

Clarity: This paper is very well-written with a good balance of technical details and general introductions. I enjoyed reading it even as someone outside of the ODE/PDE field.




**Weaknesses:**


Significance: This is probably my biggest question for the paper. What types of real-world scenarios could D-CIPHER be applied to? The Discussion section briefly mentions finding heat and vibration sources and discovering population models and epidemiological models. However, at the level of the current discussion, these applications all sound very abstract. The paper would benefit from more grounding in concrete applications. If it's possible to add in experiments on real data, that would strengthen the paper. Even if not, the paper would still be improved with a more extensive discussion on how D-CIPHER could be applied in each of the relevant scenarios. For example, what would the form be (in step 1) based on prior knowledge? How would the fields be estimate (step 2)?

**Questions:**

See "weaknesses"

**Limitations:**

The very last paragraph discusses some potential limitations, but in my opinion it would be very helpful to have some "negative examples" in the paper, i.e. synthetic experiments where D-CIPHER fails and to explore the reasons of failure.

---

> ### Author Rebuttal · Authors · 2023-08-09
>
> Dear Reviewer Znhz,
>
> We appreciate your thorough feedback and encouraging comments. We are particularly grateful for your suggestion on discussing how D-CIPHER can be applied in real-world scenarios. As a result, we have added two sections to Appendix F that discuss it in more detail. We firmly believe that these additions enhance the paper's overall quality and improve the accessibility of our approach for users.
>
> ### Actions taken
> We have added the following two sections to Appendix F:
> 1. Impact on real-world problems
> 2. D-CIPHER in practice.
>
> ### Impact on real-world problems
> D-CIPHER is especially useful in discovering governing equations for systems with more than one independent variable. For instance, spatiotemporal data or temporal data structured by age or size. In particular, we envision D-CIPHER to be useful in modeling spatiotemporal physical systems, population models, and epidemiological models.
>
> **Spatiotemporal physical systems.** D-CIPHER may prove useful in discovering equations governing the oceans or the atmosphere. For instance, some places actively add or remove CO2 from the atmosphere. These “sources” and “sinks” are likely to be described by a $\partial$-free part which D-CIPHER is specially equipped to discover. Similarly, with the ocean temperature where $\partial$-free part can describe a heat source. Another area of application can be modeling seismic waves across the earth’s crust. Here the $\partial$-free part can describe the vibration source (e.g., an earthquake).
>
> **Population models.** Population models can be used in agriculture to determine the harvest or for pest control to predict their impact on the crop. They have also been used in environmental conservation to model the population of endangered species. Population models have also been used in modeling the growth of cells to better understand tumor growth. Moreover, understanding the evolution of a population pyramid for a specific country may prove invaluable in ensuring its economic stability. As in all these scenarios, the rates of growth and mortality are likely to be described by $\partial$-free part, D-CIPHER is uniquely positioned to discover such equations as an aid to human experts.
>
> **Epidemiological models.** Epidemiological models are crucial during a pandemic for better planning and interventions. For many diseases the rates of mortality and infection are age-dependent. Thus modeling the spread of disease using PDEs (rather than ODEs) might provide superior results.
>
> ### D-CIPHER in practice
> We have added a section to Appendix F discussing guidelines and things to consider while using D-CIPHER. In particular, it discusses the following points (a more detailed discussion is provided in the appendix).
>
> **The order of the differential equation.** One of the first considerations should be to choose the order of the differential equation $K$. For many dynamical systems, $K=2$ is sufficient unless we expect very complicated behavior. Then considering $K=3$ or even $K=4$ may be warranted. Note, that we show that D-CIPHER can discover a fourth-order PDE (Kuramoto-Sivashinsky equation) in Section 8.1.
>
> **Homogeneous equations.** Before searching through the whole space of closed-form $g$ (derivative-free parts), we can consider whether the equation we want to discover may be homogeneous. These experiments on the restricted search space can provide quick insights before searching through all closed-form derivative-free parts.
>
> **Terms in a dictionary.** For a given order of a differential equation $K$, it is a good idea to include all standard differential operators up to order $K$ acting on all the variables. For instance, for $K=2$ and $M=1+1$ we could choose $\mathcal{Q} = (\partial_t u, \partial_x u, \partial_t \partial_x u, \partial_t^2  u, \partial_x^2 u)$. That allows to cover all linear PDEs with constant coefficients up to that order. To allow for non-linear PDEs we can include a term like $\partial_t{u^2} = 2 u \partial_t u$ that often describes advection (as in Burger's equation).
>
> **Dictionary steering when dealing with many dependent variables.** When we deal with a system of PDEs rather than a single PDE choosing a dictionary is increasingly important. As we explain in Appendix F.2, discovering whole systems of PDEs is very challenging and D-CIPHER is not designed to do so out of the box. However, we show how that can be done in certain situations. We can steer what kind of equations are discovered by choosing the terms in the dictionary.
>
> **Estimation algorithm.** Estimation algorithms make different assumptions on the data-generating process and should be chosen based on domain expertise. As we show in Appendix F.10, algorithms that produce smoother functions, such as Gaussian Process regression and cubic spline interpolation, tend to have good results. We can consider the advantages and disadvantages of these methods. For instance, Gaussian Process regression works very well for smooth signals. However, it is computationally intensive and might not perform well if the signal is not smooth enough (it has abrupt changes). Spline interpolation, on the other hand, is faster and more appropriate for less smooth signals, but it might introduce certain unwanted artifacts because of using cubic polynomials to interpolate the data.
>
> ### Failure modes
> We note that we not only discuss the *potential* limitations but actually show some of these limitations in our experiments. In particular, Figure 3 shows how D-CIPHER fails to discover the target equations if
> 1. The noise becomes too high
> 2. The sampling interval is too large
> 3. The number of samples is too low
>
> Thank you once again for dedicating your time to reviewing our paper. We hope our responses have been satisfactory in addressing your queries. If any aspects still require additional explanation or if you have further questions, please let us know. We are eager to provide the necessary explanations.

---

### Official Review · Reviewer_kC9e · 2023-07-24

**Soundness:** 3 good
**Presentation:** 4 excellent
**Contribution:** 3 good
**Rating:** 7
**Confidence:** 2

**Summary:**

The paper proposes a new way of discovering closed-form Partial Differential Equations (PDEs) from data. This especially aims at high-order PDEs, especially when the specific form is not pre-assumed and there is a lack of observations on derivatives. The key idea is to represent the unknown PDE with terms that are bounded by derivatives and terms that are not, so that the latter kind can be easily and reliably estimated from data, hence the ground-truth, while the first kind can be estimated by leveraging symbolic regression.
Several synthetic datasets are employed for evaluation, many of which are simulated from equations that do not satisfy the linear combination assumption made by existing work.


**Strengths:**

Strengths:

1.	A new framework for discovering PDEs, especially the ones with high-order derivatives and a lack of direct observations on the derivatives.

2.	A comprehensive evaluation on many different PDEs satisfying and beyond the assumption of PDEs made by previous work.

3.	The comparison seems to show a better performance.

4.	Good exposition. The paper has a good balance between background and technical details.


**Weaknesses:**

I am in general in favour of this paper. However, it is not my area of expertise, so it would be good if they authors could clarify some questions here.

1.	Reliance on symbolic regression.  I wonder to what extent the proposed framework has to rely on symbolic regression. This opens up several questions. (1) How inclusive or comprehensive does the dictionary has to be?  What if some key derivatives are not present in the dictionary?  (2) can the authors provide more details on the computational time in addition to E.2? It would be good to show the computation time in F.3, when the dictionary is gradually increased

2.	Comparison. It is arguably true that derivatives are hard to measure in applications. But since the experiments are done using synthetic data, I wonder what if the observations of derivatives are available? Will the proposed method still outperform existing methods? I would assume the proposed framework still works to some extent, although not being able to make use of the observed derivatives.


**Questions:**

Please see my 'Weaknesses' section above.

**Limitations:**

The authors mentioned limitations. But in real-world scenario, there are other factors that might make applying this framework difficult. The first one is the sparsity of observations. The sensors are normally not well distributed and sometimes extremely sparse. So the estimate based on the zero-order information might not be reliable to start with. The second is the type of noise, which is normally unknown and needs to be estimated with the data together.

---

> ### Author Rebuttal · Authors · 2023-08-09
>
> Dear Reviewer kC9e,
>
> We appreciate your thorough feedback and kind remarks. Your questions have undoubtedly strengthened our work. In particular, we want to thank you for your inquiry about taking advantage of the derivative data should it be available. As D-CIPHER can in fact very naturally use such additional information, we have added a section in the appendix explaining how this can be achieved. We summarize the improvements we have made to the paper based on your review and we answer your questions below.
>
> ### Actions taken
> We have added the following sections to the appendix:
> 1. Section "How comprehensive does the dictionary $\mathcal{Q}$ need to be?" in Appendix F.
> 2. Plot of the computation time as the dictionary is gradually increased in Appendix F.3
> 3. Section "How to take advantage of the observed derivatives?" in Appendix F
>
> ### Reliance on symbolic regression
> We rely on the symbolic regression algorithm to find a closed-form derivative-free part. If the equation is suspected to have a derivative-free part equal to 0 or is linear in $u$ then the symbolic regression algorithm is not necessary and we can just add the term $u$ to the dictionary $\mathcal{Q}$ (as a 0$^{\text{th}}$ order derivative). This, however, only works in scenarios when the target equation is in a linear combination form. To discover anything more complicated, we need to rely on a symbolic regression algorithm.
>
> ### How comprehensive does the dictionary $\mathcal{Q}$ need to be?
> We have attached a table (Table 1 in the attached PDF) that shows 17 different differential equations and what terms they need in a dictionary. All of them can be discovered with a dictionary of size of 10, $\mathcal{Q} = (\partial_t u, \partial_x u, \partial_t \partial_x u, \partial_t^2 u, \partial_x^2 u, \partial_t \partial_x^2 u, \partial_x^3 u, \partial_x^4 u, \partial_x(u^2), \partial_x^2 (u^2))$, and 9 of those equations can be described with a dictionary containing just 4 terms, $\mathcal{Q} = (\partial_t u, \partial_x u, \partial_t^2 u, \partial_x^2 u)$. It is thus likely that such small dictionaries are sufficient to discover most of the well-known equations. Many differential equations have similar derivative-bound parts and differ by derivative-free parts. Being able to discover *any closed-form derivative-free part* is what makes D-CIPHER stand out.
>
> ### Computation time when the dictionary is gradually increased
> As shown in Figure 1, CoLLie's computation time does not increase significantly when the dimensionality is increased. We have added in Appendix F.3 a plot that shows how the computation time increases when the dictionary is gradually increased. The plot can be seen in the attached PDF in Figure 1.
>
> ### How to take advantage of the observed derivatives
> D-CIPHER can make use of the observed derivatives (if they are available) by adapting the dictionary. Consider a setting with a dictionary $\mathcal{Q} = (\partial_t u, \partial_x u, \partial_t \partial_x u, \partial_t^2 u, \partial_x^2 u)$. If we happen to have the measurements of $\partial_t u$ then we can introduce a new variable $v = \partial_t u$ and change the dictionary to $\mathcal{Q} = (v, \partial_x u, \partial_x v, \partial_t v, \partial_x^2 u)$. Note, we have performed experiments where the dictionary contains more than one dependent variable in Appendix F.2. With observed derivatives, we can also enlarge the space of Variational-Ready PDEs by allowing $g$ ($\partial$-free part) to depend on $v$ as well.
>
> Thank you once again for dedicating your time to reviewing our paper. We hope our responses have been satisfactory in addressing your queries. If any aspects still require additional explanation or if you have further questions, please let us know. We are eager to provide the necessary explanations.

---

> > ### Comment · Reviewer_kC9e · 2023-08-17
> > **Rebuttal clarifies my questions**
> >
> > Thanks for the detailed responses and the added content. I will keep my confidence low as this is not my area of expertise but I am happy to see this paper accepted.

---

> > > ### Author Response · Authors · 2023-08-19
> > >
> > > Dear Reviewer kC9e,
> > >
> > > We appreciate your time invested in assessing our paper and the rebuttal. We are delighted to see that you reaffirmed your acceptance of our work! Your constructive comments have played a significant role in enhancing the quality of our paper.
> > >
> > > Kind regards,
> > >
> > > Authors of Submission13865

---

### Author Rebuttal · Authors · 2023-08-10

### Additional PDF

Table 1 in the attached PDF shows 17 different differential equations and what terms they need in a dictionary.

Figure 1 in the attached PDF shows how the computation time increases when the dictionary is gradually increased.

---

### Decision · Program_Chairs · 2023-09-21

**Decision:**

Accept (poster)

**Comment:**

While the subject of the paper fell slightly out of the expertise of the reviewers, they collectively appreciated the strength of the contributions of the paper. The authors' response also addressed several raised concerns. Hence, I am recommending an acceptance.